# MIRAGE: Multi-hop Reasoning with Ambiguity Evaluation for Illusory Questions

Jeonghyun Park[1]*, Ingeol Baek[1]*, Seunghyun Yoon[2], Haeun Jang[1], Aparna Garimella[2], Akriti Jain[2], Nedim Lipka[2], Hwanhee Lee[1]†

Department of Artificial Intelligence, Chung-Ang University[1], Adobe Research[2]
{tom0365, ingeolbaek, jhe020814, hwanheelee}@cau.ac.kr
{syoon, garimell, akritij, lipka}@adobe.com

## ABSTRACT

Real-world Multi-hop Question Answering (QA) often involves ambiguity that is inseparable from the reasoning process itself. This ambiguity creates a distinct challenge, where multiple reasoning paths emerge from a single question, each requiring independent resolution. Since each sub-question is ambiguous, the model must resolve ambiguity at every step. Thus, answering a single question requires handling multiple layers of ambiguity throughout the reasoning chain. We find that current Large Language Models (LLMs) struggle in this setting, typically exploring wrong reasoning paths and producing incomplete answers. To facilitate research on multi-hop ambiguity, we introduce **MultI**-hop **R**easoning with **A**mbi**G**uity **E**valuation for Illusory Questions (**MIRAGE**), a benchmark designed to analyze and evaluate this challenging intersection of ambiguity interpretation and multi-hop reasoning. MIRAGE contains 1,142 high-quality examples of ambiguous multi-hop questions, categorized under a taxonomy of syntactic, general, and semantic ambiguity, and curated through a rigorous multi-LLM verification pipeline. Our experiments reveal that even state-of-the-art models struggle on MIRAGE, confirming that resolving ambiguity combined with multi-step inference is a distinct and significant challenge. To establish a robust baseline, we propose **CL**arifying **A**mbiguity with a **R**easoning and **I**nstructi**ON** (**CLARION**), a multi-agent framework that significantly outperforms existing approaches on MIRAGE, paving the way for more adaptive and robust reasoning systems.

## 1 INTRODUCTION

Multi-hop Question Answering (QA) requires models to construct a reasoning chain by connecting information scattered across multiple documents (Trivedi et al., 2022; Yang et al., 2018; Ho et al.,

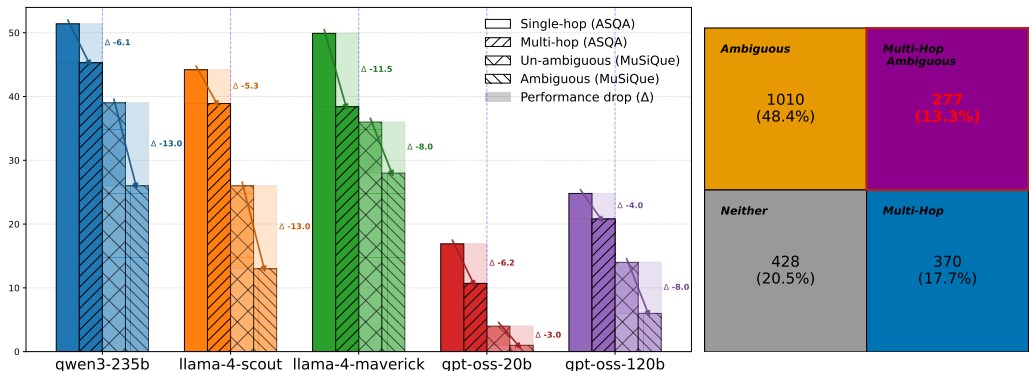

Figure 1: Performance drops under ambiguity and multi-hop (left), multi-hop ambiguity prevalence in real-world (right).

---

*Equal contribution. Authors randomized.
†Corresponding author

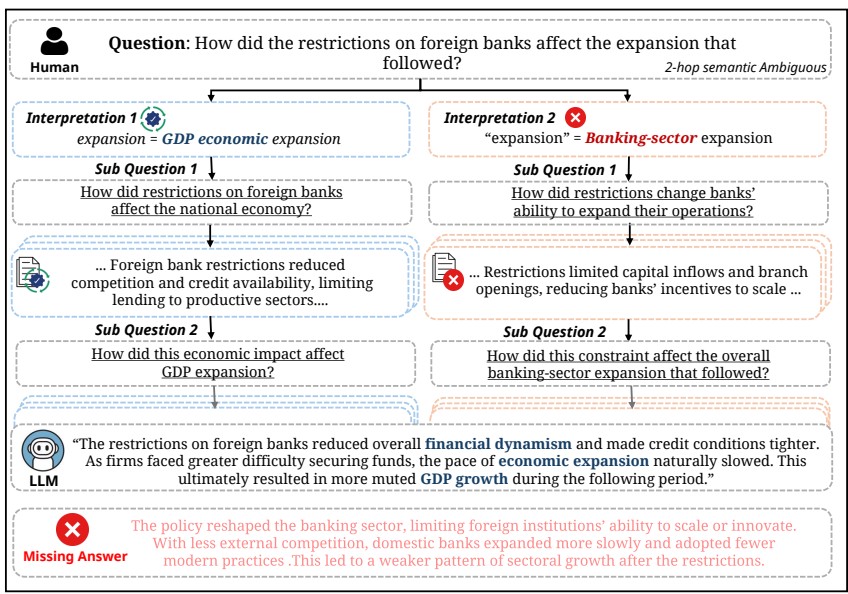

Figure 2: An example where an LLM fails to resolve a 2-hop semantic ambiguous question.

2020b). While ambiguity is a well-recognized challenge in QA (Min et al., 2020), its impact becomes even more pronounced in multi-hop settings. Unlike single-hop QA, where ambiguity can be resolved from a single context and question, multi-hop QA involves multiple plausible sub-questions, each of which may be ambiguous. This requires the QA model to perform sub-question-specific ambiguity resolution across all possible interpretations to arrive at the correct answer. Hence, the precision of the early reasoning process is critical, as errors will propagate through the entire subsequent ambiguity-resolution process. A failure at this stage can lead to a final answer that is not only incomplete but also fundamentally flawed. Previous research has underexplored this area, despite its prevalence in user conversational queries (Lee et al., 2024; Tanjim et al., 2025).

Instead, existing work has addressed these challenges along separate directions: developing multi-hop QA benchmarks that emphasize compositional reasoning without ambiguity, and creating ambiguity-focused QA benchmarks that are limited to single-hop contexts. Empirical results (Figure 1, left) show that ambiguous questions in MuSiQue (Trivedi et al., 2022) and multi-hop questions in ASQA (Stelmakh et al., 2022) both cause substantial performance degradation, underscoring the inherent difficulty of this setting. Moreover, an analysis of real-world user queries from the *lmsys-chat-1m corpus* (Zheng et al., 2024) (Figure 1, right) reveals that 48.4% of questions are ambiguous, 17.7% involve multi-hop reasoning, and 13.3% exhibit both. These findings demonstrate that ambiguity in multi-hop QA is common and unresolved, highlighting the need for benchmarks that test both reasoning and ambiguity. Moreover, when faced with multi-hop ambiguous questions, even strong LLMs tend to resolve ambiguity for only a single sub-question, pruning away alternative interpretations needed for the other hops.

For instance, as shown in Figure 2, the ambiguity in the question arises from the semantic attachment of the term "expansion": depending on whether "expansion" modifies *GDP-level economic expansion* or *banking-sector expansion*, the reasoning chain diverges into two distinct 2-hop trajectories. In the first reading, the model must assess how foreign-bank restrictions influenced the national economy and how this economic impact shaped subsequent GDP growth. In the second reading, the model must instead evaluate how those restrictions constrained banks' ability to scale and how that constraint affected the banking-sector expansion that followed. Despite these equally valid interpretations, the model commits prematurely to only one reading—or worse, produces a blended chain that merges partial evidence from both—thereby pruning the alternative interpretation required for the other hop. As each interpretation yields different sub-questions, evidence sets, and causal pathways, this collapse of interpretation-specific multi-hop reasoning leads to answers that are incomplete or missing entire branches, illustrating the inherent difficulty of a 2-hop semantically ambiguous query (see §2.1).

To address the challenges of ambiguity in multi-hop QA, we introduce **M**ult**I**-hop **R**easoning with **A**mbi**G**uity **E**valuation for Illusory Questions (**MIRAGE**), a benchmark specifically crafted to eval-

| Type | Definition | LLM Action | Typical Cues |
|---|---|---|---|
| **Syntactic** | The query can be syntactically parsed in different ways. | Resolve | Pronouns, ellipsis, PP attachment, coordination, quantifier scope. |
| **General** | The query focuses on specific information, but a broader, closely related query may better capture the user's true information need. | Generalize | Comparatives/superlatives, vague heads, "overview vs. details". |
| **Semantic** | The query may include homonyms. A word can be both a common noun and a named entity or an entity name can refer to multiple distinct entities. | Interpret | Homonyms/aliases, acronym collisions, entity-name clashes. |

Table 1: Taxonomy of multi-hop ambiguity QA, pairs definition with LLM action, detection cues.

uate multi-hop ambiguity scenarios. We construct MIRAGE from the MuSiQue dataset (Trivedi et al., 2022), which provides compositional multi-hop questions with dependency-linked hops across diverse domains. We apply a multi-LLM pipeline to detect ambiguity types in MuSiQue and filter instances by checking alignment between clarified interpretations, supporting evidence, and answers to ensure quality and objectivity.

To enable a comprehensive evaluation of this task, each of the 1,142 examples in MIRAGE provides the ambiguity type, a set of corresponding clarified questions, supporting evidence with short answers, and a long-form answer that integrates all valid interpretations.

Leveraging MIRAGE, we conduct a comprehensive empirical study to assess how well current state-of-the-art models handle multi-hop ambiguity. We find that strong LLMs systematically struggle to manage multiple reasoning paths simultaneously and often commit prematurely to a single interpretation, leading them to ignore equally valid lines of reasoning and ultimately produce incomplete or one-sided answers. These results provide strong empirical evidence that resolving ambiguity that is deeply integrated with a multi-step inference process is a distinct and largely unsolved challenge, confirming the necessity of our benchmark. To address this challenge and establish a robust baseline for the MIRAGE benchmark, we propose **CL**arifying **A**mbiguity with a **R**easoning and Instructi**ON** (**CLARION**). CLARION adopts a two-stage, multi-agent approach that clarifies ambiguity, then retrieves per interpretation, improving results on MIRAGE. Through benchmarking MIRAGE with various frameworks, we validate the unique challenges it poses and underscore the need for more adaptive, ambiguity-aware reasoning systems. In summary, (1) we introduce MIRAGE, the first benchmark for multi-hop ambiguous QA, featuring 1,142 rigorously annotated examples and a unified evaluation protocol that spans detection, clarification, retrieval, and answer generation; and (2) we propose CLARION, a multi-agent framework that establishes a solid baseline on MIRAGE.

## 2 Multi-Hop Ambiguous QA

Multi-hop ambiguous QA concerns questions that decompose into a sequence of sub-questions (hops), where each hop admits multiple plausible interpretations. Therefore, answering requires hop-wise disambiguation: at each hop, the system must choose among competing interpretations before proceeding to the next hop, with each choice conditioning subsequent retrieval and reasoning as in Figure 2. Following prior ambiguity definition (Tang et al., 2025; Tanjim et al., 2025), we classify multi-hop ambiguous QA into three types: syntactic, general, and semantic (Table 1).

### 2.1 Ambiguity Taxonomy for Multi-hop QA

To systematically analyze and address the challenges of multi-hop ambiguous QA, we extend the standard taxonomy of ambiguity into a setting where each type interacts with multi-step reasoning. Table 1 summarizes three categories—syntactic, semantic, and general—along with (i) their definitions, (ii) the reasoning actions required of an LLM (*Resolve*, *Generalize*, *Interpret*), and (iii) the types of evidence or cues that help detect and handle each category in a multi-hop QA context.

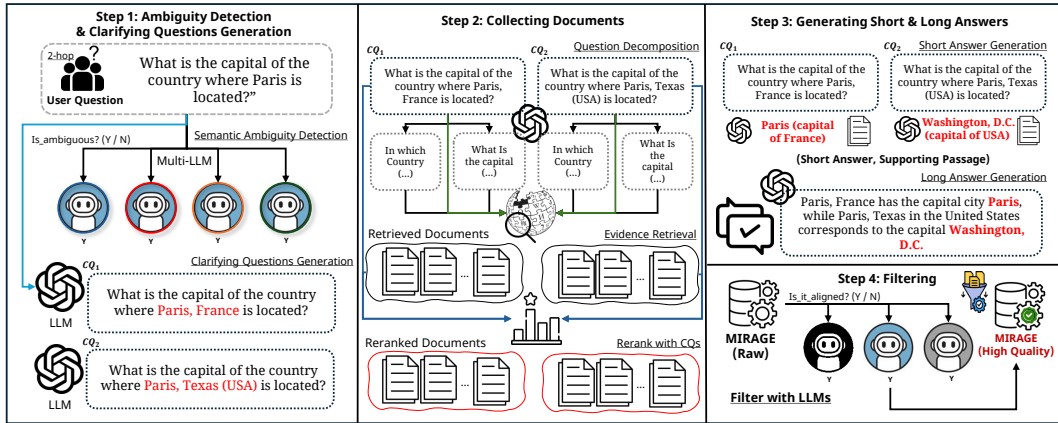

Figure 3: Overview of the four-stage MIRAGE dataset construction pipeline, which uses a multi-LLM framework to convert MusiQue into multi-hop ambiguous QA benchmark.

**Multi-hop Syntactic Ambiguity**    Multi-hop syntactic ambiguity arises when a question admits multiple well-formed syntactic parses due to uncertainty in sentence structure; each parsed question is a multi-hop question. Common triggers include prepositional phrase attachment (e.g., "with/at/in"), pronoun coreference, coordination scope ("and/or"), and quantifier scope. For example, *"I saw the man with a telescope"* permits (i) an instrument reading (the speaker used a telescope) and (ii) an attribute reading (the man had the telescope). Each interpretation leads to a different chain of reasoning, so the LLM's primary action here is RESOLVE—disambiguating the structure.

**Multi-hop General Ambiguity**    Multi-hop general ambiguity occurs when a query's scope is too narrow for the underlying information need, causing early commitment and pruning of otherwise valid hops. Typical triggers include over-specified constraints, overly fine entity/region granularity, and restrictive modifiers. Consider *"In which year did the* city center *host the Olympics?"* When the implied intent is to recover the host year(s) for the broader *metropolitan area*, where events are distributed across venues, the city-center reading prematurely narrows the plan. The correct multi-hop plan is: (i) resolve the geographic scope to the metropolitan area, (ii) enumerate Olympic events/venues in that area, and (iii) align them with host year(s). If the system commits to "city center," it prunes steps (i)–(ii) and never retrieves the necessary evidence. In this category, the model should first GENERALIZE (scope adjustment and intent refocusing). Please refer to Appendix K for detailed detection signals of multi-hop general ambiguity.

**Multi-hop Semantic Ambiguity**    Multi-hop semantic ambiguity arises when the same words map to multiple meanings or entities (polysemy, homonymy, entity collision), producing different multi-hop reasoning paths. Common triggers include ambiguous names, acronyms, and brand–common-noun overlaps. For example, given a phrase *"Apple revenue in 2012"*, the system must determine whether the intended meaning is *Apple Inc.* or the fruit. In this category, the LLM's primary action is INTERPRET (sense/entity selection): choose the intended meaning or entity using contextual cues (topic, domain, time).

## 3    MIRAGE: A BENCHMARK FOR MULTI-HOP AMBIGUOUS QA

We introduce MIRAGE, a benchmark explicitly designed to jointly evaluate ambiguity resolution and multi-hop reasoning in question answering. We process ambiguous questions from MuSiQue through four stages to build MIRAGE: (1) Ambiguity detection and clarification, which relies on a full-agreement consensus from four distinct LLMs; (2) Interpretation-specific document collection, decomposition, and re-ranking; (3) Generation of short answers for each interpretation and a final integrated long answer; and (4) A multi-LLM filtering stage to verify the alignment and quality of the instances.

### 3.1    DATASET CONSTRUCTION

We build MIRAGE from MuSiQue's validation set and a subset of its training set. Unlike recent multi-hop benchmarks (Zhu et al., 2024; He et al., 2024) that are narrow in domain or inflate hops with list-style questions (e.g., top-5), MuSiQue enforces connected, dependency-linked reasoning

| Stage | Syntactic | General | Semantic | Total |
|---|---|---|---|---|
| MuSiQue (Original) | 24,834 | 24,834 | 24,834 | – |
| After Detection & Clarification | 8,642 | 11,703 | 9,544 | 29,889 |
| After Answer Generation | 6,675 | 8,433 | 7,034 | 22,142 |
| Before Filtering | **1,239** | **1,440** | **1,651** | **4,330** |
| After Filtering (Final) | **176** | **232** | **734** | **1,142** |
| Avg. Hops | 2.95 | 2.10 | 2.54 | - |
| Avg. Question Length | 18.27 | 15.73 | 15.33 | - |

Table 2: Stage-wise drop and final statistics for **MIRAGE**. Top: samples retained after each stage. Bottom: final per-type statistics.

across diverse domains. We first filter out questions from MusiQue that lack ambiguity and retain only those judged as ambiguous by our multi-stage pipeline. Let $\mathcal{Q}_{\text{base}}$ be the set of base multi-hop questions. We consider three ambiguity types $\mathcal{T} = \{\text{Syntactic}, \text{General}, \text{Semantic}\}$. We use a set of off-the-shelf LLMs as detectors; for a question $q \in \mathcal{Q}_{\text{base}}$, type $t \in \mathcal{T}$, and detector $m$, let $y_{m,t}(q) \in \{0,1\}$ denote whether $q$ is judged ambiguous of type $t$.

**Detecting Ambiguous Questions**    Given a multi-hop user question from MuSiQue, we provide definitions of each ambiguity type and ask multiple LLMs to detect type-wise ambiguity. Because different LLMs vary in their ability to detect ambiguity, we employ four detectors: *gpt-4.1* (Achiam et al., 2023), *llama-4-maverick* (AI, 2025), *qwen3-235b-a22b* (Yang et al., 2025), and *claude-sonnet-4* (Anthropic, 2025). We keep a type label only when the detectors are *fully in agreement*.

Let $\mathcal{M}$ be the detector set ($|\mathcal{M}| = 4$). For a question $q$ and type $t \in \mathcal{T}$, detector $m \in \mathcal{M}$ outputs $y_{m,t}(q) \in \{0,1\}$. We define the full-agreement rule

$$\phi_t(q) = \mathbb{I}\left( \frac{1}{|\mathcal{M}|} \sum_{m \in \mathcal{M}} y_{m,t}(q) = 1 \right), \qquad \mathcal{T}(q) = \{\, t \in \mathcal{T} \mid \phi_t(q) = 1 \,\},$$

where $\mathcal{T}(q)$ denotes the ambiguity types assigned to $q$. Under this rule, we assign a type $t$ only if all four detectors judge $q$ ambiguous of type $t$, reducing single-model bias and yielding fair, objective, high-quality labels.

**Generating Clarified Questions**    For each $(q, t)$ with $t \in T(q)$, we use *gpt-4.1* to generate clarified questions that resolve the type-$t$ ambiguity while preserving the user's information need, denoted as $\mathcal{C}(q, t) = \{c_1, \dots, c_n\}$ with $n \geq 2$. We utilize these clarified questions as retrieval inputs.

**Collecting Documents**    To generate answers, we require evidence obtained via retrieval over clarified questions. However, clarified questions can remain multi-hop, and directly questioning with a specific multi-hop form may narrow the search scope and miss relevant documents. To mitigate this, we again use *gpt-4.1* to decompose each clarified question $c \in \mathcal{C}(q, t)$ into atomic sub-questions $\mathcal{S}(c) = \{s_1, \dots, s_k\}$. For every $s \in \mathcal{S}(c)$, we retrieve up to 10 candidate documents from English Wikipedia[1]. We then pool candidates $\mathcal{D}(c) = \bigcup_{s \in \mathcal{S}(c)} \mathcal{D}(s)$, and if $|\mathcal{D}(c)| < 10$, we perform an additional retrieval process with the clarified question $c$ itself to back-fill more evidence. Next, we perform embedding-based re-ranking with *Qwen3-8B-Embedding* (Zhang et al., 2025b) by computing similarity between the clarified question and document passages. Finally, we sort $\mathcal{D}(c)$ in descending order of this score to prioritize evidence aligned with the clarified interpretation.

**Generating Short and Long Answers**    Given each clarified question $c$ and its ranked candidate documents $\mathcal{D}(c)$, we use *gpt-4.1* to produce a short factual answer only when the retrieved evidence clearly supports it; otherwise, we omit the short answer and drop that clarified item. For retained cases, we also record the passage used for generating the short answer. Finally, for the original question $\mathcal{Q}_{\text{base}}$, we utilize *gpt-4.1* to write a single-sentence long answer that connects the two short answers into a coherent statement while preserving the clarified interpretations and pointing to their supporting passages. For the detailed prompts for construction stages, refer to Appendix L.

**Filtering**    Before filtering, we make sure short answers stay short. If either clarified question has a short answer longer than 10 tokens, we cut it down to a much shorter form by utilizing *gpt-4.1*.

---

[1]https://dumps.wikimedia.org/

After that, we remove cases where the two short answers end up being the same, so the quality of short answers is kept high. After this stringent filtering, the final MIRAGE dataset consists of **1,142** examples. For the final filtering stage, we exclude *gpt-4.1*—already used in Steps 2 and 3—and instead employ *llama-4-maverick*, *qwen3-235b-a22b*, and *claude-sonnet-4*. Each candidate instance, including the question, clarified questions, ambiguity type, supporting passages, short answers, and long answers, is independently checked for alignment in all fields. We retain only those cases where all three models unanimously judged the instance as fully aligned, using the same criteria as our human evaluation protocol. Table 2 shows statistics and reports key characteristics of MIRAGE.

## 3.2 DATASET ANALYSIS AND VALIDATION

**Ambiguity and Hop Count**    We measure the difference in average hop counts between ambiguous and unambiguous questions by extracting from the MuSiQue training set. Among these, we sample each 1,000 ambiguous and unambiguous questions. As in Table 2, the average hop count for ambiguous questions is *2.441*, while for unambiguous questions it is *2.074*. This result indicates that ambiguous questions generally involve more hops, making them inherently more challenging and underscoring the importance of addressing them effectively.

| Evaluation Protocol | Syntactic | General | Semantic |
|:---:|:---:|:---:|:---:|
| Is the user question {*type*}-ambiguous? | 83.0→94.0 (+11.0) | 88.0→92.0 (+4.0) | 93.0→94.0 (+1.0) |
| Do the clarified questions resolve the specified type of ambiguity? | 81.0→91.0 (+10.0) | 75.0→96.0 (+21.0) | 87.0→95.0 (+8.0) |
| Does the long answer include the short answers without contradiction? | 90.0→97.0 (+7.0) | 86.0→98.0 (+12.0) | 91.0→97.0 (+6.0) |
| All fields are valid | 67.0→89.0 (+22.0) | 67.0→90.0 (+23.0) | 77.0→92.0 (+15.0) |

Table 3: Human evaluation before vs. after applying the filtering method.

**Dataset Quality Assessment**    To assess dataset quality, we ask five annotators and use a majority-vote scheme. We first sample 20 instances per ambiguity type (60 total) and obtain binary (YES/NO) judgments for each item in our protocol, yielding 300 judgments in total (5 annotators × 60 items). As shown in Table 3, annotators evaluate whether (i) the question exhibits ambiguity of the specified type, (ii) the clarified question resolves the original ambiguity, and (iii) the generated long answer contains the corresponding short answers. We repeat the same evaluation after our final filtering on a new 60-instance sample (again 300 judgments) and observe consistent improvements across all criteria. For criterion (i), the average YES rate across the three types increases from **88.0%** (pre-filtering) to **93.3%** (post-filtering), a gain of **+5.3** percentage points. For criterion (ii), it rises from **81.0%** to **94.0%** (**+13.0**). For criterion (iii), it increases from **89.0%** to **97.3%** (**+8.3**). These results support the effectiveness of our filtering pipeline and the overall quality of MIRAGE for evaluating multi-hop ambiguity. For the details about the information of the annotator and inter-agreement about them, please refer to Appendix D

**Distribution of Ambiguity Types**    As shown in Table 2, syntactic types are often discarded since different parses collapse into identical short answers or show inconsistent type assignments after clarification. General cases are frequently too broad, leading to alignment failures, so these two types are filtered more heavily, and the final dataset is skewed toward semantic ambiguity.

## 4   CLARION: AN AGENTIC FRAMEWORK FOR MULTI-HOP AMBIGUOUS QA

To effectively tackle the multi-hop QA, we propose **CLARION** (CLarifying Ambiguity with a Reasoning and InstructiON), a two-stage agentic framework designed to systematically detect, disambiguate, and answer multi-hop ambiguous questions. CLARION mitigates these failures by separating the task into two distinct phases: **Planning** and **Acting** as outlined in Figure 4.

**Planning Agent**    The *Planning Agent* serves as a planning module that analyzes the input question before any retrieval or answering. It performs three sequential operations: (1) **Ambiguity Detection**:

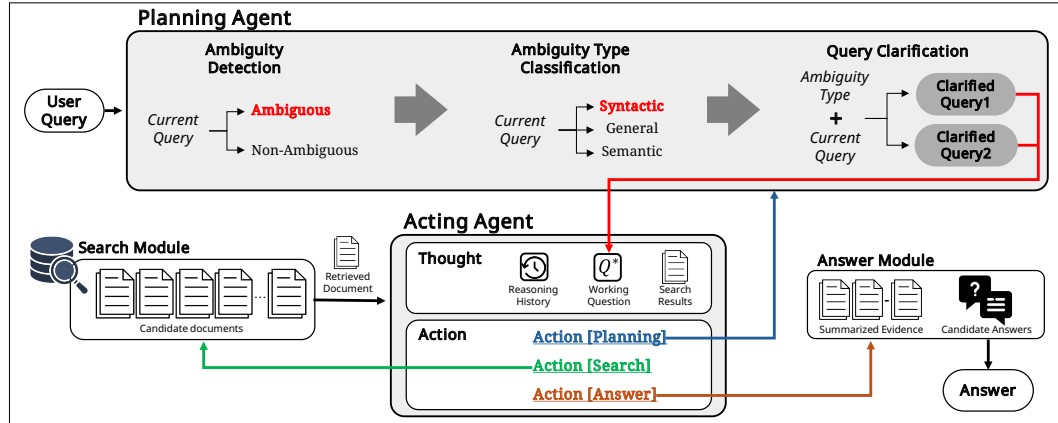

Figure 4: Overview of our CLARION framework. A *Planning Agent* resolves ambiguity, and an *Acting Agent* executes a ReAct loop to generate the final answer.

the agent determines whether the question contains ambiguity. If the question is unambiguous, it is immediately passed to the *Acting Agent*. (2) **Ambiguity Type Classification**: if ambiguity is detected, the question is categorized into one of three predefined types: *Syntactic*, *General*, or *Semantic*. (3) **Question Clarification**: based on the detected type, the agent rewrites the original question into clarified variants that resolve the ambiguity while preserving the information need. These clarified questions constitute the execution plan for downstream reasoning.

**Acting Agent**    The *Acting Agent* executes the reasoning plan through a *ReAct-style prompting* (Yao et al., 2023) scheme, unfolding in a **Thought → Action → Observation** loop. At each iteration, the agent selects one of three actions: (1) **Search**: retrieve external documents when additional evidence is needed; (2) **Planning**: re-invoke detection, type classification, or clarification if the current plan is insufficient; (3) **Answer**: synthesize the final output once enough evidence has been gathered. To ensure reliable parsing and automated execution, all actions generated by the agent must be in JSON format. Furthermore, to prevent infinite loops and ensure computational tractability, the ReAct prompting is limited to a maximum of five iterations. If this limit is reached without a resolution, the agent is compelled to execute the *Answer* action, formulating the best possible response based on the information gathered thus far. We provide Implementation details and the full prompt templates used at each stage in Appendix H and Appendix L, respectively.

## 5  EXPERIMENTS AND RESULTS

We benchmark MIRAGE using three standard baselines for solving multi-hop and ambiguous QA under open domain QA settings and our proposed CLARION. We first describe the setup, then report overall results (Table 4), followed by an ablation that isolates the contribution of *Detection* and *Clarification*, and additional analysis.

**Models**    We utilize three strong LLMs, *qwen3-235b-a22b-2507* (Yang et al., 2025), *gemini-2.5-flash* (Comanici et al., 2025), and *deepseek-chat-v3.1* (Liu et al., 2024), which are widely used in real-world scenarios to assess how these practical models handle MIRAGE. Because both multi-hop and ambiguous QA fall under the open-domain QA setting, solving them requires access to external knowledge sources. Thus, we adopt search-based baselines as our general solution paradigm. All baselines share the same embedding model for retrieval (*qwen3-embedding-8b*) and identical retrieval hyperparameters (fixed top-$k$), ensuring a fair and consistent comparison across models.

**Evaluation metrics**    We use three metrics that evaluate the model's long and short answers. *STR-EM* measures explicit coverage between the model's long answer and the set of gold short answers. After simple text normalization, it checks what fraction of gold short answers are literally contained in the model output. *Disambig-F1* measures extractability between the model's long answer and each clarified question. A frozen extractive QA scorer reads the model's long answer as context and tries to extract an answer for each clarified question. We compute token-level F1 against the gold short answers and average. *LLM-as-a-Judge* (Gu et al., 2024) measures long answer quality between

| Model | Method | STR-EM (%) | Disambig-F1 (%) | Avg (%) | LLM-as-a-Judge | Latency (s) |
|---|---|---|---|---|---|---|
| | No Retrieval | 20.98 | 21.19 | 21.09 | 3.083 | 0.439 |
| | CoT | 21.51 | 22.32 | 21.91 | 2.897 | 1.295 |
| | NaiveRAG | 25.10 | 26.20 | 25.65 | 2.752 | 0.667 |
| | CoT w/ RAG | 25.63 | 26.61 | 26.12 | 2.947 | 1.042 |
| Qwen3-235b | DIVA | 28.82 | 22.73 | 25.78 | 3.015 | 1.026 |
| | ReAct | 20.98 | 21.00 | 20.99 | 2.832 | 3.820 |
| | CLARION (Ours) | **38.73** | **28.38** | **33.56** | **3.474** | 8.958 |
| | CLARION w/o clarification | 25.10 | 25.56 | 25.33 | 2.922 | 4.968 |
| | CLARION w/o clarification & detection | 22.94 | 24.02 | 23.48 | 2.782 | 4.443 |
| | No Retrieval | 15.59 | 20.10 | 17.85 | 2.307 | 0.169 |
| | CoT | 16.32 | 17.52 | 16.92 | 2.258 | 0.476 |
| | NaiveRAG | 22.16 | **28.63** | 25.40 | 2.297 | 0.220 |
| | CoT w/ RAG | 23.15 | 27.31 | 25.23 | 2.373 | 0.440 |
| Gemini-2.5 | DIVA | 18.82 | 20.29 | 19.56 | 2.303 | 0.506 |
| | ReAct | 21.32 | 22.37 | 21.84 | 2.428 | 2.290 |
| | CLARION (Ours) | **29.12** | 26.30 | **27.71** | **2.752** | 2.576 |
| | CLARION w/o clarification | 24.12 | 22.54 | 23.33 | 2.609 | 2.629 |
| | CLARION w/o clarification & detection | 23.82 | 22.04 | 22.93 | 2.573 | 2.567 |
| | No Retrieval | 17.75 | 18.72 | 18.24 | 2.683 | 0.226 |
| | CoT | 19.80 | 22.12 | 20.96 | 2.512 | 0.862 |
| | NaiveRAG | 20.20 | 25.03 | 22.62 | 2.084 | 0.246 |
| | CoT w/ RAG | 21.33 | 23.18 | 22.25 | 2.632 | 0.778 |
| DeepSeek-v3.1 | DIVA | 18.82 | 20.66 | 19.74 | 2.636 | 0.746 |
| | ReAct | 23.17 | 24.78 | 23.97 | 2.723 | 4.167 |
| | CLARION (Ours) | **31.47** | **27.03** | **29.25** | **3.042** | 5.566 |
| | CLARION w/o clarification | 23.63 | 22.99 | 23.31 | 2.927 | 4.863 |
| | CLARION w/o clarification & detection | 24.51 | 24.31 | 24.41 | 2.906 | 4.116 |

Table 4: **MIRAGE** benchmark results with baseline reasoning methods. Metrics are scaled to percentages. Best per-model scores for each metric are in **bold**, and second-best scores are underlined.

the predicted and gold long answers under the original user question. A judge assigns LLM scores for Relevance, Faithfulness, Informativeness, and Correctness (0–5), and we report the average score.

**Baselines**     **(1) No Retrieval**: LLM-only inference without any external context. We directly prompt the model to answer accurately and concisely. **(2) CoT** (Wei et al., 2022): A Chain-Of-Thought prompting baseline that encourages step-by-step reasoning for the ambiguous question without using retrieved documents. **(3) NaiveRAG** (Lewis et al., 2020): A standard retrieve-then-read pipeline that encodes the original question, retrieves top-$k$ passages, and generates an answer conditioned on the retrieved contexts. We instruct the model to rely on passages when they clearly contain the answer and otherwise fall back to background knowledge. **(4) CoT with RAG** (Wei et al., 2022; Lewis et al., 2020): An extension of CoT where the model is given the retrieved passages and asked to perform chain-of-thought reasoning grounded in the retrieved evidence. **(5) DIVA** (In et al., 2025): A three-stage *diversify–verify–adapt* RAG framework for ambiguous questions. DIVA rewrites the question into several concrete interpretations, retrieves evidence independently for each, deduplicates passages to collect complementary evidence, and labels passages as Useful / Partial-Useful / Useless to adapt the answering strategy. **(6) ReAct** (Yao et al., 2023): A reasoning–acting prompting baseline where the model alternates natural-language *Thought* steps with retrieval *Action* calls, using the retrieved observations to iteratively refine its reasoning and produce the final answer.

**Main Results**     As shown in Table 4, our proposed CLARION consistently outperforms both LLM-only and RAG-based baselines on MIRAGE. Ablation studies further confirm the contribution of each component: removing either *Detection* or *Clarification* lowers performance, with the largest drop observed when *Clarification* is omitted. This indicates that identifying ambiguity and explicitly rewriting the question to cover plausible interpretations is crucial for aligning retrieval with user intent and for robust multi-hop ambiguous QA.

**Performance by Ambiguity Type**     To understand how different types of ambiguity affect model behavior, we further break down CLARION's performance by ambiguity type. Table 5 reports STR-EM, Disambig-F1, and LLM-as-a-Judge scores for each ambiguity category (Semantic, Syntactic, General) across all three backbone models. Across models, we observe a consistent pattern: instances annotated as *General* ambiguity achieve the highest scores, followed by *Syntactic* and then *Semantic* ambiguity. This ordering holds for all three metrics (STR-EM, Disambig-F1, and LLM-

as-a-Judge), suggesting that semantic ambiguities are the most challenging cases, whereas general under-specification is relatively easier for current LLMs to resolve.

| Model | Ambiguity Type | STR-EM | Disambig-F1 | LLM-as-a-Judge |
|---|---|---|---|---|
| *Qwen3-235b* | Syntactic | 35.00 | 25.81 | 3.370 |
| | General | 46.45 | 33.91 | 3.860 |
| | Semantic | 33.90 | 24.86 | 3.303 |
| *Gemini-2.5* | Syntactic | 29.67 | 26.53 | 2.652 |
| | General | 32.79 | 28.71 | 3.059 |
| | Semantic | 24.86 | 23.61 | 2.637 |
| *DeepSeek-v3.1* | Syntactic | 29.33 | 27.17 | 2.840 |
| | General | 34.43 | 28.78 | 3.316 |
| | Semantic | 30.23 | 25.11 | 2.940 |

Table 5: Performance by ambiguity type on MIRAGE. Metrics for STR-EM and Disambig-F1 are scaled to percentages. Across all three backbone models, General ambiguity cases yield the highest scores, followed by Syntactic and then Semantic ambiguity.

**Validating the LLM Judge**  We validate the use of the *LLM judge* for our main experiments by comparing its 0–5 scores with human ratings on 300 items across four criteria. We measure (i) linear association with Pearson $r$, (ii) rank consistency with Spearman $\rho$ and Kendall $\tau_b$, and (iii) grade-level agreement on the 0–5 scale via Quadratic Weighted Kappa (QWK). As shown in Figure 5, we observe consistently strong correlations across all families; QWK further indicates grade-aligned agreement, supporting the LLM judge as a valid proxy for our main experiments. For full details on the human evaluation protocol, see Appendix J.

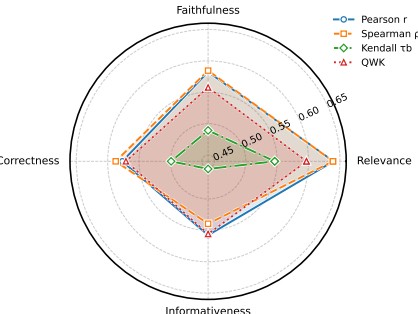

Figure 5: Correlation between LLM and human judgments.

## 6  RELATED WORK

**Ambiguity in QA**  Ambiguity in Open Domain QA arises when a question is polysemous or context-dependent, permitting multiple reasonable interpretations. In such ambiguous QA settings, systems should either (i) transform the user's input into a set of disambiguated questions that collectively cover the plausible readings (question clarification (Min et al., 2020)), or (ii) pose targeted clarifying questions to get the user's intent (asking clarifying questions (Zhang & Choi, 2025; Zhang et al., 2025a)), and, in either case, (iii) produce long-form answers that explicitly address each interpretation (long-form QA (Stelmakh et al., 2022)). AmbigQA (Min et al., 2020) has formalized this problem setting and introduced both a dataset and an evaluation protocol. In the long-form QA paradigm, ASQA (Stelmakh et al., 2022) explicitly requires the long answer that synthesizes multiple plausible interpretations of an ambiguous question into a single, comprehensive response. SituatedQA (Zhang & Choi, 2021) systematizes context-dependent ambiguity by conditioning answers on external situational factors (e.g., time and location). More recently, AmbigDocs (Lee et al., 2024) have been proposed that deliberately include ambiguous entity names and provide mutually conflicting evidence sets, evaluating whether models can resolve entity-level ambiguity and produce internally consistent answers. To address ambiguity, Many recent RAG-based pipelines (Tanjim et al., 2025) extend the workflow by first disambiguating the ambiguous question and then answering each clarified question. Tree of Clarifications (Kim et al., 2023) augments this process with external retrieval, expanding interpretations along a branching structure. These approaches do fine on single-hop cases, but they fall short on multi-hop questions where each step adds more ambiguity. DIVA (In et al., 2025) introduces a diversify–verify–adapt framework that rewrites ambiguous questions into concrete interpretations, retrieves evidence for each, and synthesizes a comprehensive

answer by integrating diverse supporting passages. DIVA is effective for single-hop ambiguity, but it is not designed to resolve ambiguity and reasoning dependencies in multi-hop settings.

| Benchmark | Scale | Tasks | Multi-hop? | Short Ans.? | Long Ans.? | Ambig. Type Diversity? |
|---|---|---|---|---|---|---|
| AmbigQA | 14,042 | Ambiguous QA | ✗ | ✓ | ✗ | ✗ |
| CAMBIGNQ | 5,653 | Ambiguity Detection, Clarifying Question Generation | ✗ | ✓ | ✗ | ✗ |
| CondAmbigQA | 200 | Conditional Ambiguous QA | ✗ | ✓ | ✗ | ✗ |
| ASQA | 5,301 | Long-form QA | ✗ | ✓ | ✓ | ✗ |
| AmbigDocs | 36,098 | Ambiguous QA | ✓ | ✓ | ✗ | ✗ |
| **MIRAGE (Ours)** | 1,142 | **Multi-hop Ambiguity Detection**, **Multi-hop Clarifying Question Generation**, **Multi-hop Long-form QA** | ✓ | ✓ | ✓ | ✓ |

Table 6: Comparison of ambiguous QA benchmarks and MIRAGE.

**Multi-hop QA**    Multi-hop QA requires extracting and connecting evidence from multiple documents (He et al., 2024; Zhu et al., 2024; Tang & Yang, 2024). HotpotQA (Yang et al., 2018) designs multi-hop by using Wikipedia-based questions that prompt models to retrieve one or more articles and explicitly sentence-level supporting facts. These questions demand reasoning that spans across multiple sources. 2WikiMultihopQA (Ho et al., 2020a) refines multi-hop explainability by providing both structured and unstructured evidence along with reasoning paths. MuSiQue (Trivedi et al., 2022) constructs problems where "connected reasoning" is essential by composing single-hop questions that depend on one another. It underscores that many multi-hop datasets can be solved by shallow shortcuts that skip real composition, making MuSiQue more challenging. Unlike prior ambiguous QA benchmarks such as AmbigQA (Min et al., 2020), ASQA (Stelmakh et al., 2022), CAMBIGNQ (Lee et al., 2023), and CondAmbigQA (Li et al., 2025), which primarily target ambiguity in single-hop questions, MIRAGE explicitly addresses ambiguity in a multi-hop setting. It is accompanied by a robust baseline (see Section 4) for systematic assessment and future development. In addition, MIRAGE provides a richer set of subtasks spanning multi-hop ambiguity detection, clarification, and QA. As summarized in Table 6, this design introduces a new challenge for ambiguous QA by requiring models to resolve and reason over ambiguity that spans multiple hops.

## 7    CONCLUSION

We introduce MIRAGE, a benchmark that explicitly targets ambiguity in multi-hop QA. We release 1,142 carefully annotated questions with type-specific clarifications, evidence-backed short/long answers, and supporting passages. We further provide a unified evaluation protocol that jointly assesses ambiguity detection, clarification, retrieval, and answer generation within a single framework. Finally, we propose CLARION, a robust baseline on MIRAGE, showing that powerful models struggle when ambiguity and multi-hop reasoning coincide.

**Ethics Statement**    MIRAGE benchmark is constructed entirely from publicly available data sources (MuSiQue, Wikipedia), ensuring that no personally identifiable or private information is present. We use a multi-LLM consensus pipeline for ambiguity detection and filtering, reducing the risk of individual model bias or hallucination. Expert contributors with their consent conduct all human annotation, and no unfair labor practices are involved. While our dataset and evaluation pipeline strive to minimize bias, users should be aware that language models may still inherit subtle biases from the underlying data. We encourage responsible use and further analysis of potential risks when applying MIRAGE or derived models in real-world settings.

**Reproducibility Statement**    To ensure reproducibility, we release the entire MIRAGE dataset and all experimental settings, including prompt templates and evaluation scripts. Detailed descriptions of the construction pipeline and model configurations are provided in the main paper. All LLMs and retrieval tools used in this study are publicly available or accessible via APIs.

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

## A  THE USE OF LARGE LANGUAGE MODELS

We write the manuscript ourselves and an LLM (ChatGPT-5) is used solely for refinement—style, clarity, and grammar. It is not used for ideation or content generation.

## B  LIMITATION AND FUTURE STEPS

**Dataset Balance and Augmentation**  While MIRAGE covers all three ambiguity types, the current dataset has fewer general and syntactic ambiguity cases than semantic ones. This distribution reflects real-world multi-hop queries as sampled from MuSiQue, but future work could augment and further balance these categories to facilitate more fine-grained analysis and targeted method development.

**CLARION Framework Complexity**  CLARION demonstrates strong performance but introduces additional system complexity due to its multi-agent structure and planning-acting cycle. Although this complexity enables richer ambiguity resolution, future work could explore lighter-weight or more efficient designs without sacrificing effectiveness, making deployment and integration even more practical.

**Toward More Robust Solutions**  Despite CLARION's effectiveness, MIRAGE surfaces the ongoing difficulty of achieving complete and faithful resolution for all multi-hop ambiguous queries. Our results showcase both the progress and the remaining gaps in current methods, providing a solid foundation and clear direction for continued innovation in this important area.

## C  DOMAIN DIVERSITY OF MIRAGE

We tag the topic of each question using *gpt-oss-120b* and report the distribution in Table 7. As shown in Table 7, MIRAGE covers a broad range of subject areas. The most frequent topics include *"History"*, *"Geography&Places"*, and *"Politics&Government"*, indicating diverse coverage beyond any single domain.

| Domain | Ratio |
|---|---|
| Science & Technology | 2.79 |
| Math & Logic | 0.07 |
| History | **30.88** |
| Geography & Places | 20.44 |
| Politics & Government | 10.16 |
| Business & Economics | 2.49 |
| Society & Culture | 1.76 |
| Arts & Literature | 4.32 |
| Entertainment (Film/TV/Games) | 8.59 |
| Music | 7.48 |
| Sports | 6.44 |
| Religion & Philosophy | 2.31 |
| Medicine & Health | 0.18 |
| Nature & Environment | 1.57 |
| UNKNOWN | 0.51 |
| **Total** | **100%** |

Table 7: MIRAGE domain coverage.

## D HUMAN ANNOTATOR DETAILS

We employ five graduate-level annotators fluent in English for all labeling tasks in our study. Annotators were compensated at a rate of 10 USD per hour. Each annotator received detailed guidelines and example cases before annotation, and ambiguous cases were discussed through controlled calibration sessions. Reporting inter-annotator agreement is crucial for assessing the reliability of human judgments. Therefore, we compute Fleiss' $\kappa$, average category-wise agreement ($\bar{P}$), strict agreement (all annotators selecting the same label), and majority agreement (at least three annotators agreeing) for both (1) long-answer judgment dimensions (Relevance, Faithfulness, Informativeness, Correctness) and (2) dataset quality evaluation dimensions (Ambiguity, Clarification, Long Answer).

Table 8 summarizes the results. Overall, annotators exhibit consistently high agreement across evaluation criteria. The relatively low $\kappa$ value for the *Long Answer* quality dimension stems from a well-known prevalence effect: when nearly all annotators overwhelmingly choose the same label (here, "Yes"), Fleiss' $\kappa$ is distorted downward due to artificially inflated chance agreement. Importantly, the strict and majority agreement rates for this item remain very high, confirming that annotators were indeed consistent and that the low $\kappa$ does *not* reflect genuine disagreement.

| Item | Fleiss' $\kappa$ | $\bar{P}$ | Strict Agree | Maj. Agree |
|------|-------|------|------|------|
| Relevance | 1.000 | 1.000 | 1.000 | 1.000 |
| Faithfulness | 1.000 | 1.000 | 1.000 | 1.000 |
| Informativeness | 0.907 | 0.978 | 0.967 | 0.989 |
| Correctness | 1.000 | 1.000 | 1.000 | 1.000 |
| Ambiguity | 0.589 | 0.978 | 0.967 | 0.989 |
| Clarification | 0.851 | 0.989 | 0.983 | 0.994 |
| Long Answer | -0.006 | 0.989 | 0.983 | 0.994 |

Table 8: Inter-annotator agreement across long-answer judgments and dataset quality evaluation. We report Fleiss' $\kappa$, average agreement $\bar{P}$, strict agreement, and majority agreement.

## E ASQA BENCHMARK RESULTS

As shown in Table 9, in ASQA, our agentic approach consistently delivers the strongest short-answer performance across models. With detection + clarification enabled, CLARION achieves the best per-model averages—e.g., Qwen3-235b-a22b-250: 71.64 vs. 62.24 (NaiveRAG) and 55.54 (DIVA); Gemini-2.5-Flash: 68.81 vs. 58.08 and 51.08; DeepSeek-Chat-v3.1: 69.48 vs. 51.34 and 50.32. Improvements appear in both STR-EM (coverage of gold short answers) and Disambig-F1 (extractability for clarified questions), indicating that explicitly detecting ambiguity and rewriting the query steers retrieval to interpretation-aligned evidence rather than memorized or mixed contexts. Ablations verify the contribution of each component, with the largest drop when clarification is removed—highlighting that planning for ambiguity before acting is crucial even on single-hop-oriented datasets like ASQA. Together, these trends support our claim that agentic planning and acting modules are broadly beneficial beyond MIRAGE and strengthen answer completeness and precision in ambiguous QA settings.

## F ANALYSIS ON CLARIFICATION STEPS

To analyze why *clarification* yields substantial performance gains, we measure token-level F1 for each sample by comparing (i) answers generated from the original user question before clarification (BASE) and (ii) answers generated from clarified queries (CLAR1, CLAR2) against the dataset's gold answers. Figure 6 visualizes the per-query slope of $\Delta F1 = F1_{\text{CLAR}} - F1_{\text{BASE}}$, showing that the average F1 after clarification consistently exceeds the BASE condition. This gain arises because the clarification step resolves ambiguity in the original question, steering retrieval toward evidence passages that directly support the correct interpretation. Consequently, the LLM produces more accurate and complete answers, indicating that clarification is not a mere paraphrase but an effective disambiguation-driven retrieval mechanism that significantly improves F1.

| Model | Method | Short Answer | | |
|---|---|---|---|---|
| | | **STR-EM** | **Disambig-F1** | **Avg** |
| *Qwen3-235b-a22b-250* | No Retrieval | 70.98 | 36.56 | 53.77 |
| | NaiveRAG | 83.14 | 41.34 | 62.24 |
| | DIVA | 73.53 | 37.54 | 55.54 |
| | CLARION *(ours)* | **92.94** | **50.33** | **71.64** |
| | CLARION *w/o clarification* | 85.69 | 42.58 | 64.13 |
| | CLARION *w/o clarification & detection* | 84.31 | 42.30 | 63.30 |
| *Gemini-2.5-Flash* | No Retrieval | 66.86 | 34.79 | 50.82 |
| | NaiveRAG | 76.86 | 39.30 | 58.08 |
| | DIVA | 66.47 | 35.69 | 51.08 |
| | CLARION *(ours)* | **89.61** | **48.02** | **68.81** |
| | CLARION *w/o clarification* | 88.43 | 46.09 | 67.26 |
| | CLARION *w/o clarification & detection* | 87.65 | 46.04 | 66.84 |
| *DeepSeek-Chat-v3.1* | No Retrieval | 70.59 | 35.57 | 53.08 |
| | NaiveRAG | 67.84 | 34.85 | 51.34 |
| | DIVA | 75.10 | 25.54 | 50.32 |
| | CLARION *(ours)* | **90.98** | **47.99** | **69.48** |
| | CLARION *w/o clarification* | 87.45 | 46.16 | 66.81 |
| | CLARION *w/o clarification & detection* | 88.82 | 46.23 | 67.53 |

Table 9: ASQA results across methods and models. We report STR-EM / Disambig-F1 in %. Best per model in **bold**, second-best underlined.

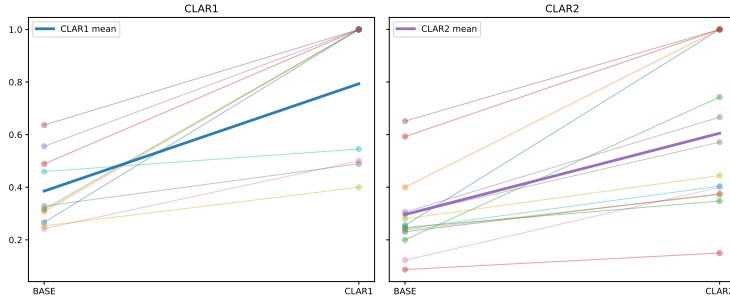

Figure 6: Overview of the MIRAGE Construction Process.

## G CASE STUDY

### G.1 MIRAGE CONSTRUCTION

In this subsection, we present a short case study of meaningful instances that are removed during the final filtering stage, as shown in Table 10. These cases are dropped because, after shortening the short answers, both clarified interpretations collapsed into the same string. Although the rule keeps answer quality high, it also removes valid ambiguous cases.

**Syntactic** Two clarified questions differ in grammatical structure ("in the birthplace of" vs. "in the place where ... was born"), but both shorten to the same short answer. **General** Two clarified questions ask about different ways of defining Antarctica's border, yet both reduce to the same numeric answer. **Semantic** Two clarified questions focus on different semantic aspects (actor identity vs. character role), but shortening collapsed both to the same name.

| Type | QID | Clarified Query 1 | Clarified Query 2 | SA$_1$ | SA$_2$ |
|------|-----|-------------------|-------------------|--------|--------|
| Syntactic | 4hop1_8294 15324_26424 _581618 | Who founded the chain of music-themed restaurants whose first establishment was located in the birthplace of the person who ejected the Benedictines in 1559? | Who founded the chain of music-themed restaurants with its first establishment in the place where the person who ejected the Benedictines in 1559 was born? | Isaac Tigrett | Isaac Tigrett |
| General | 2hop__100274 _14948 | What latitude marks the northern border of Antarctica? | At what latitude is the continental boundary of Antarctica defined? | 60° S | 60° S |
| Semantic | 2hop__725611 _52870 | Which actor from *Michael Collins* appears in *The Phantom Menace*, and which character do they portray? | In *The Phantom Menace*, which character is played by an actor who was also in *Michael Collins*? | Liam Neeson | Liam Neeson |

Table 10: **Failure cases during MIRAGE construction due to short-answer shortening collisions.** After shortening, both short answers collapsed to identical strings, causing removal even though the clarified queries represent distinct interpretations.

## G.2 Failure Cases of CLARION

Despite its strong performance, CLARION still fails on certain ambiguous multi-hop queries. Table 11 shows representative failure cases across all three ambiguity types. In each case, CLARION's prediction collapses to a single interpretation without resolving ambiguity, so the system generates only one short answer and misses the gold interpretations. This results in complete mismatches (0 on STR-EM and Disambig-F1). These errors illustrate how mis-specified sub-questions or over-broad interpretations derail reasoning and retrieval, leading to a complete mismatch against gold answers.

| Type | Original Query | Predicted Long Answer | Gold Long Answer | Fail Reason |
|------|----------------|----------------------|------------------|-------------|
| Semantic | What city shares a border with the place where the person who went to the state known for its Mediterranean climate during the gold rush worked? | Stockton | Brooklyn and Traverse City share borders with the relevant places. | Collapsed to a single interpretation; failed to clarify multiple possible places. |
| Syntactic | Who won the Indy Car Race in the largest populated city of the state where Yuma's Library District is located? | Mario Andretti (Phoenix, 1993) | Álex Palou and Hélio Castroneves | Relied on historical fact lookup; failed to disambiguate event scope and multiple winners. |
| General | Who brought the language Hokkien to the country on the natural boundary between the country that hosted the tournament and the country where A Don is from? | The Hoklo (Hokkien) people | Dutch colonial administration and Hokkien-speaking immigrants during Spanish colonization | Did not resolve broad/general query; simplified to one actor instead of multiple sources. |

Table 11: **Representative CLARION failure cases.** CLARION often fails to clarify ambiguous sub-questions and collapses to a single short answer, leading to zero scores on both STR-EM and Disambig-F1.

## H Implementation Details

As shown in Table 12, we report our implementation details for our MIRAGE construction pipeline and running CLARION.

**API Access and Infrastructure** All LLM calls were made using the OpenRouter API. Experiments were run on a workstation equipped with an RTX 6000 Ada GPU.

**Models** We use four off-the-shelf LLMs as ambiguity detectors: GPT-4.1 (OpenAI), Llama-4-Maverick (Meta), Qwen3-235b (Alibaba), and Claude-Sonnet-4 (Anthropic). For generating clarified questions and answers, we exclusively use GPT-4.1. For LLM-as-a-judge filtering, we employ Llama-4-Maverick, Qwen3-235b, and Claude-Sonnet-4.

**Decoding and Prompting** All LLM calls (for both detection and generation) were run with temperature set to 0.0 and a maximum token limit of 512. All prompts and task templates are described in detail in Appendix L.

**Retrieval Pipeline** For evidence retrieval, we use FAISS for fast vector search over Wikipedia passages. Query and document embeddings are computed with the Qwen3-8B-Embedding model. Retrieval is performed with a fixed top-$k$ of 10 per query.

| Item | Value / Setting |
|---|---|
| API | OpenRouter API |
| Detection Model | GPT-4.1, Llama-4-Maverick, Qwen3-235b, Claude-Sonnet-4 |
| Generator Model | GPT-4.1 |
| Filtering Model | Llama-4-Maverick, Qwen3-235b, Claude-Sonnet-4 |
| LLM-as-a-Judge Model | GPT-4.1 |
| Temperature | 0.0 (detection), 0.0 (generation) |
| Max Tokens | 512 (detection), 512 (generation) |
| Evaluation Protocol | See Appendix J and Table 3 |
| Embedding Model | Qwen3-8B-Embedding |
| Retriever | FAISS |
| Top-k | 10 |
| Agent Max Search Iteration | 5 |
| GPU | RTX 6000 Ada |

Table 12: Implementation details.

**Agentic Reasoning** For CLARION, the agent's maximum search iteration is set to 5. The planning agent performs ambiguity detection, type classification, and clarification as described in Section 4; the acting agent executes search and answer steps up to the iteration limit.

**Filtering and Evaluation Protocol** After answer generation, candidate instances are filtered using three LLMs (excluding GPT-4.1 to prevent overfitting). An instance is retained only if all models unanimously judged every field (question, clarifications, type, evidence, answers) as fully aligned, following the same protocol as human evaluation (see Appendix J and Table 3 for details).

# I   LLM JUDGE-DETAILED SCORE

| Method | Model | LLM-as-a-Judge (0–5) | | | |
|---|---|---|---|---|---|
| | | Relevance | Faithfulness | Informativeness | Correctness |
| *LLM-only* | | | | | |
| No Retrieval | *Qwen-3-235b* | 3.324 | 3.147 | 2.931 | 2.931 |
| | *Gemini-2.5* | 2.643 | 2.439 | 1.912 | 2.233 |
| | *DeepSeek-v3.1* | 3.067 | 2.706 | 2.480 | 2.480 |
| *RAG-based baselines* | | | | | |
| Naive RAG | *Qwen-3-235b* | 2.961 | 2.988 | 2.357 | 2.829 |
| | *Gemini-2.5* | 2.525 | 2.620 | 1.643 | 2.543 |
| | *DeepSeek-v3.1* | 2.325 | 2.408 | 1.516 | 2.259 |
| Diva | *Qwen-3-235b* | 3.073 | 3.465 | 2.565 | 3.084 |
| | *Gemini-2.5* | 2.531 | 2.694 | 1.727 | 2.382 |
| | *DeepSeek-v3.1* | 2.918 | 2.851 | 2.292 | 2.596 |
| *CLARION (ours)* | | | | | |
| CLARION *(w/o clarification & detection)* | *Qwen-3-235b* | 3.057 | 2.957 | 2.496 | 2.696 |
| | *Gemini-2.5* | 2.673 | 2.963 | 2.039 | 2.702 |
| | *DeepSeek-v3.1* | 3.075 | 3.159 | 2.635 | 2.839 |
| CLARION *(w/o clarification)* | *Qwen-3-235b* | 3.184 | 3.084 | 2.608 | 2.890 |
| | *Gemini-2.5* | 2.712 | 2.963 | 2.133 | 2.700 |
| | *DeepSeek-v3.1* | 3.089 | 3.136 | 2.699 | 2.843 |
| CLARION | *Qwen-3-235b* | 3.600 | 3.551 | 3.502 | 3.271 |
| | *Gemini-2.5* | 2.843 | 3.106 | 2.302 | 2.824 |
| | *DeepSeek-v3.1* | 3.228 | 3.177 | 2.943 | 2.882 |

Table 13: LLM-as-a-Judge sub-criteria. All scores are on a 0–5 scale.

As shown in Table 13, we report the scores of each sub-criterion under the LLM-as-a-Judge evaluation for the baselines and for CLARION. Consistent with our main experiments, CLARION generally achieves higher judge scores across most criteria.

## J    HUMAN EVALUATION PROTOCOLS

| Criterion | Description |
| --- | --- |
| Relevance | Does the long answer fully address both clarified queries and include all relevant short answers, without digression? |
| Faithfulness | Is the answer consistent with the intent and facts in the original query, clarified queries, and short answers? |
| Informativeness | Does the answer provide additional useful background, explanations, or actionable guidance to fulfill the user's needs? |
| Correctness | Are all facts accurate, with no errors or omissions in the key information? |

Table 14: Human evaluation protocol for long answer quality. Used for correlation analysis in Figure 5.

For the correlation analysis presented in Figure 5, we develop a dedicated human evaluation protocol to systematically assess long answer quality. Annotators rate each answer using the detailed criteria shown in Table 14, with explicit written instructions for each aspect. This protocol is introduced exclusively for the evaluation setup in Figure 5, ensuring that all human judgments are directly comparable with the figure's correlation metrics.

## K    DETECTION CUE FOR GENERAL AMBIGUITY

General ambiguity arises when a query is over-specified (e.g., exact dates, versions, quoted spans) so that retrieval narrows the user's true intent. We use three complementary signals. First, *total_hits* $H(q)$ flags abnormally small result sets, indicating a narrowed scope. Second, the *KL divergence* $D_{\text{KL}}(P_{\text{top}} \parallel P_{\text{corpus}})$ measures how skewed the top-snippet word distribution is relative to the background corpus, revealing over-reliance on special tokens (dates, numbers, quoted phrases). Third, the *relax_delta_ratio* $\rho(q) = \frac{H(\text{relax}(q))}{H(q)}$ is an intervention-style cue: it asks how much the hit count jumps when we remove exactly one constraint (a date, a number, or a quoted span). In combination, $H(q)$ low, $D_{\text{KL}}$ high, and $\rho(q)$ high strongly suggest over-specialization–induced recall failure, whereas low $H(q)$ with low $\rho(q)$ points to genuinely sparse topics rather than over-specification. These cues reduce false positives and guide the LLM toward expert, evidence-aware judgments.

## L    PROMPT TEMPLATES

This section summarizes the prompt templates used to construct MIRAGE. For each ambiguity type, we provide templates for detection, clarification, answer generation (short/long), and query decomposition from Figures 7 to 15.

---

You are a linguistics expert.

1) Read the sentence below.
2) Decide whether it is syntactically ambiguous under any of the 18 phenomena.
3) If ambiguous, list all applicable phenomenon numbers (ascending).

**Phenomena (1–18)**

1. PP Attachment (including instrument vs. attribute "with"); 2. Relative-Clause Attachment; 3. Coordination Scope (and/or); 4. Comparative Attachment / Ellipsis; 5. Quantifier / Negation Scope; 6. Dangling / Misplaced Modifier; 7. Genitive-Chain Attachment; 8. Complement vs. Adjunct; 9. Gerund vs. Participle; 10. Ellipsis / Gapping; 11. If-clause Attachment; 12. Right-Node Raising; 13. Adjective Stacking / Coordination; 14. Inclusive vs. Exclusive "or"; 15. Adverbial Attachment (VP vs. S); 16. Focus / Only-scope; 17. Apposition vs. Restriction; 18. Degree / Comparative subdeletion.

**Question**: QUESTION

**Output (JSON)**: "is_ambiguous": "Y", "categories": [1, 3, 7]  // [] if "N"

Keys must be exactly "is_ambiguous" and "categories". No extra text.

---

*Figure 7: Prompt template for syntactic ambiguity detection.*

---

You are a linguistics expert.

The question below is syntactically ambiguous. Write at least MIN_VERSIONS distinct clarified questions, each encoding a different structural reading (attachment, scope, etc.). Preserve the topic; each rewrite must be fully unambiguous; concise, natural English.

**Question**: QUESTION

**Output (JSON)**: "clarified_queries": ["...", "..."]

Key exactly "clarified_queries"; provide at least 2 strings; no other keys.

---

*Figure 8: Prompt template for syntactic clarification.*

---

You are a linguistics expert.

1) Read the search query and three RAW metric values.
2) Decide if the query shows general ambiguity (over-specific constraints harming recall).
3) Output ONLY the JSON object in the required format.

A query with general ambiguity (over-specific) is narrowly constrained (dates, version numbers, quoted strings, etc.), likely missing the broader intent.

**Metrics**

---

**Total_hits**: Result count for the literal query.
**KL_divergence**: D_KL between top-k snippet unigrams and the whole corpus.
**Relax_delta_ratio**: Largest fold-increase in hits after removing one numeric/date/quoted constraint.

**Question**: QUESTION

**Raw metric values**

**Total_hits**: TOTAL_HITS
**KL_divergence**: KL_DIVERGENCE
**Relax_delta_ratio**: RELAX_DELTA_RATIO

**Output (JSON)**: "is_ambiguous": "Y"  // "N" if not general

Use expertise; no hard thresholds. No markdown, code fences, or extra keys.

*Figure 9: Prompt template for general (over-specific) ambiguity detection.*

You are an information-retrieval and linguistics expert.

Rewrite the query below into at least MIN_VERSIONS broader, faithful variants that surface the user's core intent and remove needless specificity or indirections.

**How to clarify**

1) Identify the core question (fact or relationship truly sought).
2) Resolve or drop cascading indirections (replace "the country where X was born" with the direct entity if obvious; else use a neutral placeholder).
3) Remove or soften excessive constraints (exact dates, versions, quoted titles).
4) Keep the answer type the same; do not over-broaden. Write concise English.

**Question**: QUESTION

**Output (JSON)**: "clarified_queries": ["...", "..."]

Key must be exactly "clarified_queries"; provide at least 2 strings; no extra keys.

*Figure 10: Prompt template for general clarification.*

You are a linguistics expert.

Semantically ambiguous lacks sufficient context so multiple reasonable meanings or referents are possible (unclear pronoun, vague time, polysemy, etc.).

1) Read the sentence.
2) Output "Y" if semantically ambiguous, else "N".
**Question**: QUESTION

**Output (JSON)**: "is_ambiguous": "Y"  // "N" if unambiguous

Key must be exactly "is_ambiguous". No extra text.

*Figure 11: Prompt template for semantic ambiguity detection.*

You are a linguistics expert.

Rewrite the semantically ambiguous question into at least 2 distinct clarified questions, each resolving a different interpretation. Preserve the original topic. Add only minimal context (time, referent, sense) to make each unambiguous. Concise English.

**Question**: QUESTION

**Output (JSON)**: "clarified_queries": ["...", "..."]

Key exactly at least 2 "clarified_queries"; no other keys.

*Figure 12: Prompt template for semantic clarification.*

You are an extractive QA assistant.

Given a question and one passage, return the shortest exact span in the passage that answers the question. If no answer, return an empty string.

**Question**: QUESTION

**Passage**: PASSAGE

**Output (JSON)**: "short_answer": "..."

Extractive only (verbatim span); no justification or extra text.

*Figure 13: Prompt template for short answer generation (extractive).*

You are an expert open-domain QA assistant.

Combine two validated short answers (A1, A2) to create a single, coherent long answer to the original ambiguous question (OQ). If both can be true, merge into 1–3 fluent sentences. Do not invent facts beyond A1 and A2.

Return only JSON that matches the schema: SCHEMA

**Question**: QUESTION

**Clarified Q1 — Short Answer A1**

CQ1
A1 = A1

**Clarified Q2 — Short Answer A2**

CQ2
A2 = A2

**Output (JSON)**: "long_answer": "..."

*Figure 14: Prompt template for long answer generation (merge A1 + A2).*

You are an information-retrieval expert.

Break the complex question into the minimal set of atomic, single-hop sub-questions in the exact order needed to fully answer it.

- Output each sub-question as a Markdown bullet starting with "* ".
- Each sub-question must ask for exactly one fact or relationship.
- No explanations or extra text.
**Question**: QUESTION

**Output (JSON)**: "sub_query": "..."

*Figure 15: Prompt template for query decomposition (ordered single-hop bullets).*

You are an expert at analyzing query ambiguity.
Your task is to determine whether a query is ambiguous and to classify the ambiguity type.

Analyze the following query and decide:

1. Provide brief reasoning.
2. Is the query ambiguous?
3. Which specific aspects make it ambiguous?
4. What extra information would clarify it?
5. Classify the ambiguity as one of: *"syntactic"*, *"general"*, *"semantic"*, or *"none"*.

**Definitions**:

* **syntactic**: multiple plausible grammatical parses (attachment/scope/coordination/pronoun reference).
* **general**: over-specific query where a broader, closely related formulation better matches the user's need.
* **semantic**: syntax is clear but meaning/intent admits multiple valid interpretations via world knowledge.

**Query**: {query}

**Return STRICT JSON**:

"reasoning": "string",
"is_ambiguous": true/false,
"ambiguity_type": "syntactic" | "general" | "semantic" | "none",
"ambiguous_aspects": ["..."],
"clarification_needed": "string"

*Figure 16: Prompt template for ambiguity detection and typing (strict JSON).*

You are an expert at clarifying ambiguous queries.
Given the original query and an ambiguity analysis, rewrite the query into **two** specific, actionable, and faithful clarified versions.

**Original Query**: {query}
**Ambiguity Analysis (JSON)**: {analysis}

**Write STRICT JSON**:

"reasoning": "why these clarifications resolve the ambiguity",
"clarified_query1": "string",
"clarified_query2": "string"

*Figure 17: Prompt template for generating two clarified queries from an ambiguity analysis.*

You are a research assistant following ReAct (Reasoning, Acting, Observing).

**Available Actions**:

* `SEARCH[query]` → run a search using the configured method
* `ANSWER[planning]` → run a planning agent
* `ANSWER[text]` → provide a final answer now

**Constraints**:

* Max searches allowed: max_searches
* Searches used so far: current_searches
* Do **not** reuse the exact same search query as previously used in context.

**Task Query**: {query}
**Previous Context**:
{context}

**Instructions**:

1. THINK about the next best step.
2. If more evidence is needed, choose `SEARCH[very specific query]`.
3. If sufficient, choose `ANSWER[concise, well-supported answer]`.
4. If you have already reached the maximum allowed searches, you **must** output `ANSWER[...]` now.

Respond in **EXACT** format:
```
THOUGHT: <your internal reasoning, one short paragraph>
ACTION: SEARCH[...specific query...] OR
ACTION: PLANNING[...call planning agent...]  OR
ACTION: ANSWER[...final answer...]
```

*Figure 18: Prompt template for ReAct-style retrieval and answering with a bounded search budget.*

