# OpenReview forum: "MIRAGE: Multi-hop Reasoning with Ambiguity Evaluation for Illusory Questions"
_ICLR.cc/2026/Conference — ICLR 2026 Conference Withdrawn Submission_

### Official Review · Reviewer_vvS5 · 2025-10-28

**Soundness:** 2
**Presentation:** 2
**Contribution:** 1
**Rating:** 2
**Confidence:** 3

**Summary:**

This paper introduces MIRAGE, a benchmark targeting the intersection of multi-hop reasoning and ambiguity, and proposes CLARION, a two-stage agentic framework (planning→acting) that first detects/classifies/clarifies ambiguity and then performs ReAct-style retrieval and synthesis. MIRAGE (1,142 examples) provides type labels (syntactic/general/semantic), clarification pairs, supporting passages, short answers, and integrated long answers, built via multi-LLM consensus and stringent filtering. Experiments across strong LLMs show sizable degradation on multi-hop ambiguity and consistent gains from CLARION, especially due to clarification.

**Strengths:**

1. The paper crisply formulates multi-hop ambiguity with a clear taxonomy (syntactic/general/semantic) and aligns each type to actionable LLM behaviors, then validates it with a rigorously constructed benchmark (multi-LLM consensus, evidence-backed short/long answers).

2. The CLARION framework operationalizes “clarify-before-retrieve” via a planning→acting pipeline and demonstrates consistent, meaningful gains over strong baselines, isolating clarification as the key driver through careful ablations.

**Weaknesses:**

1. The benchmark’s “multi-hop + ambiguity” is synthesized from MuSiQue/Wikipedia, yielding tidy linguistic ambiguities that underrepresent real tasks. In practical settings (e.g., diagnosing a GitHub repo issue), ambiguity spans versions, environments, conflicting reports, and evolving evidence across issues/PRs/logs—far beyond entity sense/parse choices. Thus MIRAGE likely misestimates real complexity and offers limited guidance for authentic, high-noise workflows.

2. CLARION’s pipeline (detect→type→clarify→ReAct) mirrors MIRAGE’s annotation/evaluation protocol, inviting prior bias. The paper lacks transfer tests to non-Wikipedia, multi-source, or tool-using scenarios (e.g., code, tables, logs), and no user-study/human-judged deployment evidence. Without cross-domain validation or robustness to noisy/contradictory evidence, claims of broader utility remain unsubstantiated.

3. The paper does not compare against a minimal prompt-driven agent loop that many practitioners would try first. For example, a single-model ReAct-style baseline with a concise system prompt such as: “You are an ambiguity-aware, retrieval-assisted QA model. For any question, enumerate plausible interpretations, retrieve evidence per branch, verify coverage, and produce a consolidated long answer. Be precise and avoid hallucinations.” This simple setup may capture much of the claimed benefit without multi-agent orchestration and needs to be included for comparison.

4. As a benchmark paper, the evaluated model set is narrow. It omits stronger reasoning-centric models/variants (e.g., reasoning-tuned or tool-use–optimized LLMs), which might already handle multi-hop ambiguity well. Without broader model coverage, it is hard to assess whether MIRAGE is genuinely challenging or merely exposes gaps in the specific models tested.

**Questions:**

See Weaknesses.

---

> ### Author Response · Authors · 2025-11-27
>
> **W1 – Realism and applicability of the benchmark.** \
> MIRAGE is designed specifically as a benchmark for linguistic multi-hop QA ambiguity. As illustrated in Figure 1, the types of queries captured by MIRAGE closely resemble those that users frequently ask in **real-world interactions with LLMs**, making the benchmark highly realistic and practically relevant.
>
> **W2 – Is CLARION tailored only to MIRAGE?** \
> We would like to stress that CLARION  is built on a broadly applicable pattern: detecting ambiguity, enumerating plausible interpretations, and running independent retrieval and verification loops for each interpretation. Rather than relying on a fixed, dataset-specific decomposition, CLARION **adopts an agentic structure** in which a Planning Agent and an Acting Agent interact to gradually resolve ambiguity over the course of reasoning. This makes it an intuitive and general framework that can be applied beyond MIRAGE rather than a hand-crafted rule system tied to a particular benchmark. Moreover, as information sources broaden, additional interpretive factors inevitably emerge, and the nature of ambiguity can change. For this work, we therefore deliberately **restrict the retrieval pool** so that we can precisely measure how effectively a model can identify, disentangle, and resolve ambiguity given a **fixed knowledge source**. This experimental design is aligned with MIRAGE’s core motivation—to isolate and rigorously evaluate ambiguity-resolution capability itself—and other reviewers have agreed with this focus. Finally, as stated in Appendix C, because **MuSiQue (the source dataset for MIRAGE) already spans diverse topics** (people, science, geography, history, etc.), we believe CLARION’s approach reflects ambiguity-resolution processes that naturally arise in a wide range of realistic scenarios, rather than being confined to a narrow question template.
>
> **W3 - Prompting-based baselines.** \
> We follow the reviewer’s suggested pipeline—identifying possible interpretations, retrieving evidence for each, and validating the final answer—through the DIVA baseline. The process the reviewer describes (enumerating all plausible interpretations for a question → performing retrieval for each branch → conducting final verification) corresponds directly to the approach proposed in the DIVA paper. Our work already adopts DIVA as a baseline, and thus this interpretation-wise retrieval and verification pipeline is incorporated into our experimental comparisons. We conducted additional experiments using ReAct and Chain-of-Thought (CoT) prompting, and CoT with RAG,  and the results are shown in [TABLE A] below.
>
> [TABLE A]
> | Model           | Method      | STR-EM (%) | Disambig-F1 (%) | LLM-as-a-Judge |
> |----------------|------------|:------------:|:------------------:|:----------------:|
> | *Qwen3-235b*   | CoT        | 21.51      | 22.32            | 2.897          |
> | *Qwen3-235b*   | CoT w/ RAG | 25.63     | 26.61            | 2.947          |
> | *Qwen3-235b*   | ReAct      | 20.98      | 21.00            | 2.832          |
> | *Gemini-2.5*   | CoT        | 16.32      | 17.52            | 2.258          |
> | *Gemini-2.5*   | CoT w/ RAG | 23.15      | 27.31            | 2.373         |
> | *Gemini-2.5*   | ReAct      | 21.32      | 22.37            | 2.428          |
> | *DeepSeek-v3.1*| CoT        | 19.80      | 22.12            | 2.512          |
> | *DeepSeek-v3.1*| CoT w/ RAG | 21.33      | 23.18            | 2.632          |
> | *DeepSeek-v3.1*| ReAct      | 23.17     | 24.78            | 2.723          |
>
> While some ReAct and CoT configurations outperform the Naive RAG results in Table 4, others fall short. This inconsistency suggests that ambiguity resolution relying solely on structured prompting remains fundamentally limited. In contrast, our agentic system, CLARION, provides a more robust and effective approach to handling multi-path ambiguity.
>
> **W4) – Limited model scope.** \
> As shown in Table 4, we additionally evaluated **Gemini-2.5 and DeepSeek-v3.1**. For Gemini-2.5, the model size is not publicly available because it is a closed-source model. Furthermore, we excluded the GPT series used during dataset creation from evaluation in order to avoid potential self-preference bias [1].
>
> [1] LLM Evaluators Recognize and Favor Their Own Generations, Neurips 2024

---

> > ### Comment · Reviewer_vvS5 · 2025-11-27
> > **Response to Rebuttal**
> >
> > Thanks for the response. However, the major concerns I clearly stated in the original review remain unaddressed even with the extra content provided in the response. Therefore, I will maintain the current score.

---

### Official Review · Reviewer_jmuS · 2025-11-01

**Soundness:** 3
**Presentation:** 2
**Contribution:** 2
**Rating:** 4
**Confidence:** 3

**Summary:**

This paper introduces MIRAGE, a rigorously constructed benchmark targeting multi-hop question answering under ambiguity, and provides the CLARION agentic baseline for systematic ambiguity resolution in open-domain multi-hop QA.  MIRAGE is derived via a multi-stage pipeline involving four distinct LLMs, resulting in 1,142 annotated examples with diverse ambiguity types, explicit clarification steps, and both short- and long-form answers. Comprehensive experiments show that modern LLMs underperform on MIRAGE, highlighting the intrinsic challenge of ambiguity in multi-hop QA.

**Strengths:**

1. MIRAGE is the first benchmark to focus explicitly on ambiguity in multi-hop QA

2. The authors propose CLARION, a multi-agent framework, as a robust baseline to tackle the MIRAGE benchmark. This provides a starting point for future research

3. The results report evaluation over a wide range of state-of-the-art LLMs and retrieval-augmented baselines.

**Weaknesses:**

1. As shown in Table 2, the dataset skews heavily toward semantic ambiguity, making the current evaluation results also rely on the ability to deal with the semantic ambiguity. Besides, it seems that the authors do not provide details results for each category.

2. The dataset size is small, and whether we can use the dataset to increase the ability to deal with the ambiguity with such a small size is questionable. Besides, the dataset is constructed by filtering and processing a single existing dataset. This reliance on one source might limit the diversity of domains.

3. The benchmark's construction and quality control pipeline is heavily dependent on LLMs. Though the human evaluation is included, the details of the human evaluation are missing, including who did the evaluation, and 60 samples are not very large.

4. Though the dataset is constructed for multi-hop QA, there is no special ambiguity type for the multi-hop setting. It seems like the dataset captures ambiguity within questions but not ambiguity that arises across reasoning hops, such as dependency conflicts or path fusion errors.  Besides, the paper doesn't clearly distinguish its dataset's challenge from a simpler single-hop ambiguity problem. In detail, the experiments show that models like GPT-4 do poorly on the new MIRAGE dataset (the hard, multi-hop ambiguous questions). But the paper never shows that these same models do well on the easy (single-hop) version of these questions. For example, it doesn't prove that GPT-4 can correctly answer a simple, single-hop question like "Where is Paris?" but then fails on the multi-hop version. Because this simple experiment is missing, we don't know if the models are failing because of the multi-hop reasoning (the paper's claim) or just because they were always bad at handling that "Paris" ambiguity in the first place.

**Questions:**

Please refer to the weaknesses part.

---

> ### Author Response · Authors · 2025-11-27
>
> **W1,2 – Dataset imbalance and size.** \
> As shown in Table 2, we initially extracted approximately 4,330 ambiguous questions from the full MuSiQue dataset. The dataset size was then substantially reduced through several filtering stages: removing items with identical short answers, eliminating cases where LLMs produced inconsistent ambiguity classifications, and discarding instances that failed our alignment criteria. We selected MuSiQue as the base source because it is a large, high-quality multi-hop dataset that spans a wide range of domains. As demonstrated in Appendix C, MIRAGE indeed **covers diverse domains derived from this foundation**. Additionally, conducting further experiments directly on MuSiQue would allow researchers to obtain complementary samples that reflect other ambiguity types or distributions not included in MIRAGE.
>
> **W3 - MIRAGE Construction details.** \
> We included detailed information about the human annotators in the revised paper. While synthetic data generation using LLMs has become common, our pipeline adds multiple layers of filtering to ensure that MIRAGE remains a **high-quality benchmark**. Five **graduate-level annotators fluent in English** contributed to the labeling, each compensated at **10 USD per hour**. Inter-annotator agreement was assessed using Fleiss’ κ, average pairwise agreement (P̄), strict agreement, and majority agreement across both long-answer evaluation criteria and dataset-quality criteria below:
> | Item           | Fleiss’ κ | P̄    | Strict Agree | Majority Agree |
> |----------------|:-----------:|:-------:|:--------------:|:----------------:|
> | Relevance      | 1.000     | 1.000 | 1.000        | 1.000          |
> | Faithfulness   | 1.000     | 1.000 | 1.000        | 1.000          |
> | Informativeness| 0.907     | 0.978 | 0.967        | 0.989          |
> | Correctness    | 1.000     | 1.000 | 1.000        | 1.000          |
> | Ambiguity      | 0.589     | 0.978 | 0.967        | 0.989          |
> | Clarification  | 0.851     | 0.989 | 0.983        | 0.994          |
> | Long Answer    | -0.006    | 0.989 | 0.983        | 0.994          |
>
> The results show uniformly **high strict and majority agreement, indicating strong consistency among annotators**. The low κ observed for the Long Answer category is a known statistical artifact caused by extreme class imbalance and does not indicate actual disagreement. Overall, these results confirm that the annotation process is reliable and robust. About the sample of human annotators, this task is inherently demanding. Our pilot study showed that annotating 60 samples required approximately five hours per annotator due to the need to inspect the original ambiguous question, verify the type of ambiguity, check whether the clarification steps genuinely disambiguate the question, and ensure that the long answer faithfully integrates all short answers. Given the time and cost required per sample, we determined that the current number of annotated instances represents a realistic and appropriate scale within the scope of this work.
>
>
> **W4 – Evidence from ASQA vs. MIRAGE.** \
>  As shown in Table 7 of the Appendix, we also conducted experiments on ASQA, a **single-hop ambiguous dataset** that is less challenging than MIRAGE. Both the baselines and CLARION achieved **stronger performance on ASQA**, indicating that models **handle single-hop ambiguity relatively well**. In contrast, the substantial performance drop on MIRAGE demonstrates that adding a multi-hop requirement significantly increases difficulty, making it much harder for models to generate accurate answers under ambiguity.

---

> > ### Comment · Reviewer_jmuS · 2025-11-27
> >
> > Thanks for the author's reply. I will adjust my score accordingly.

---

### Official Review · Reviewer_tRyh · 2025-11-01

**Soundness:** 2
**Presentation:** 3
**Contribution:** 2
**Rating:** 4
**Confidence:** 3

**Summary:**

This paper presents MIRAGE, a new benchmark for evaluating large language models (LLMs) on multi-hop reasoning under ambiguity. The dataset contains over 1K carefully curated instances where each question exhibits syntactic, general, or semantic ambiguity, accompanied by clarified sub-questions, supporting passages, short answers, and an integrated long answer. MIRAGE is constructed through a four stage multi-LLM pipeline involving ambiguity detection, clarification, document retrieval, answer generation, and cross model alignment filtering to ensure high quality annotation. The authors further introduce CLARION, a two stage multi-agent framework combining a Planning Agent for ambiguity detection and clarification with an Acting Agent performing ReAct-style reasoning to search and synthesize final answers. Experiments across strong LLMs demonstrate that CLARION outperforms RAG-based and LLM-only baselines on three complementary metrics and LLM-as-a-Judge, showing the benefit of explicit ambiguity modeling.

**Strengths:**

1.	Proposes a benchmark for evaluating ambiguity-aware multi-hop reasoning.
2.	Introduces a baseline framework integrating planning and reasoning agents.
3.	Reveals consistent limitations of current LLMs in handling ambiguous multi-hop questions.

**Weaknesses:**

1.	Limited model scope.
The analysis of LLM limitations is based only on mid-sized open models (Qwen3-235B, LLaMA-4 variants, GPT-OSS-20B/120B), without including top-tier models such as GPT-4/5 or Gemini-1.5, making it unclear whether the observed weaknesses generalize to current sota systems.
2.	Narrow baseline coverage.
The comparison focuses mainly on No-Retrieval, NaiveRAG, and DIVA, omitting several relevant categories of baselines such as (i) ambiguity-focused QA frameworks (e.g., AmbigDocs, AmbigQA), (ii) compositional reasoning agents (e.g., ReAct, Self-Ask, CoT+RAG), and (iii) ambiguity-aware retrieval methods (e.g., AmbiRAG, TACO). Including representatives from these families would offer a more comprehensive evaluation of CLARION’s relative effectiveness.
3.	Dataset imbalance.
As noted in Appendix B, the MIRAGE dataset is heavily skewed toward semantic ambiguity, with far fewer examples of syntactic or general ambiguity. This imbalance limits the benchmark’s representativeness and its ability to test model robustness across ambiguity types.
4.	High system complexity and limited practicality.
CLARION introduces a multi-agent, multi-stage pipeline (Planning → Acting) that improves accuracy but also increases computational cost and implementation overhead. Its scalability and real-world applicability remain untested.
5.	Moderate methodological novelty.
While the task formulation is new, the CLARION framework mainly recombines existing paradigms. Its reasoning loop closely follows ReAct, and its clarification stage resembles DIVA’s multi-interpretation rewriting. The innovation lies more in system integration and benchmarking than in novel algorithmic techniques.

**Questions:**

1.	Do the authors observe similar limitations in ambiguity-aware multi-hop reasoning even for frontier models such as GPT-4o or GPT-5? If so, could they provide evidence or discussion on whether the observed weaknesses generalize beyond mid-sized open models?
2.	Could the authors add or report results for additional baselines, such as ambiguity-focused QA frameworks (e.g., AmbigDocs, AmbigQA) or reasoning agents (e.g., ReAct, Self-Ask, CoT + RAG)? It would be valuable to see how CLARION compares to these stronger or more specialized methods.
3.	What is the complexity–performance trade-off of CLARION? Specifically, how does the additional planning and acting loop affect token consumption and runtime cost, relative to the performance improvements shown in Table 4?
4.	Since the MIRAGE dataset is heavily skewed toward semantic ambiguity, overall accuracy may not fully reflect model robustness. Could the authors report per-type performance (syntactic / general / semantic) to show whether CLARION performs consistently across ambiguity categories?

---

> ### Author Response · Authors · 2025-11-27
>
> **W1) – Limited model scope.**
>
> As shown in Table 4, we additionally evaluated **Gemini-2.5 and DeepSeek-v3.1**. For Gemini-2.5, the model size is not publicly available because it is a closed-source model. Furthermore, we excluded the GPT series used during dataset creation from evaluation in order to avoid potential self-preference bias [1].
>
> [1] LLM Evaluators Recognize and Favor Their Own Generations, Neurips 2024
>
> **W2 (Q2) – Additional baselines.**
>
> We conducted additional experiments using ReAct and Chain-of-Thought (CoT) prompting, and CoT with RAG, and the results are shown in [TABLE A] below.
>
> [TABLE A]
> | Model           | Method      | STR-EM (%) | Disambig-F1 (%) | LLM-as-a-Judge |
> |----------------|------------|:------------:|:------------------:|:----------------:|
> | *Qwen3-235b*   | CoT        | 21.51      | 22.32            | 2.897          |
> | *Qwen3-235b*   | CoT w/ RAG | 25.63     | 26.61            | 2.947          |
> | *Qwen3-235b*   | ReAct      | 20.98      | 21.00            | 2.832          |
> | *Gemini-2.5*   | CoT        | 16.32      | 17.52            | 2.258          |
> | *Gemini-2.5*   | CoT w/ RAG | 23.15      | 27.31            | 2.373         |
> | *Gemini-2.5*   | ReAct      | 21.32      | 22.37            | 2.428          |
> | *DeepSeek-v3.1*| CoT        | 19.80      | 22.12            | 2.512          |
> | *DeepSeek-v3.1*| CoT w/ RAG | 21.33      | 23.18            | 2.632          |
> | *DeepSeek-v3.1*| ReAct      | 23.17     | 24.78            | 2.723          |
>
> While some ReAct and CoT configurations outperform the Naive RAG results in Table 4, others fall short. This inconsistency suggests that ambiguity resolution relying solely on structured prompting remains fundamentally limited. In contrast, our agentic system, CLARION, provides a more robust and effective approach to handling multi-path ambiguity.
>
> **W3) – Dataset imbalance.** \
> We constructed MIRAGE by randomly sampling 24,834 examples from the MuSiQue dataset and then releasing all remaining instances that passed our dataset-generation pipeline. During this process, we applied a rigorous multi-stage filtering procedure: we removed trivial or pervasive ambiguity patterns and relied on LLM-based alignment checks to **retain only high-quality, well-formed ambiguous questions**. This yielded a curated subset with **high annotation and ambiguity quality**. Additionally, if one conducts experiments directly on the full MuSiQue dataset, it is possible to obtain complementary samples that reflect other types or distributions of ambiguity beyond those emphasized in MIRAGE.
>
> **W4 (Q3) Complexity of CLARION.** \
> The Planning phase of CLARION generates on average 2–3 concretized questions, and the ReAct loop in the Acting phase is limited to a maximum of three iterations.
>
> |Table B|
>
> | Model          | No Retrieval | NaiveRAG | DIVA  | CoT   | CoT w/RAG | ReAct | CLARION (Ours) | CLARION w/o clarification | CLARION w/o clarification & detection |
> |----------------|:------------:|:-------:|:-----:|:-----:|:---------:|:-----:|:---------------:|:-------------------------:|:----------------------------------:|
> | Qwen-235b      |    0.439     |  0.667  | 1.026 | 1.295 |   1.042   | 3.820 |      8.958      |           4.968           |             4.443                  |
> | Gemini-2.5     |    0.169     |  0.220  | 0.506 | 0.476 |   0.440   | 2.290 |      2.576      |           2.629           |             2.567                  |
> | DeepSeek-v3.1  |    0.226     |  0.246  | 0.746 | 0.862 |   0.778   | 4.167 |      5.566      |           4.863           |             4.116                  |
>
> According to the results in Table B, our agentic system is more complex than standard Naive RAG and ReAct approaches; however, the performance results in Table 4 show that the CLARION method is an effective way to resolve ambiguous queries.
>
> **W5) – Primary contribution and methodological positioning.** \
> The primary contribution of our work is the **MIRAGE benchmark dataset**. We provide a high-quality benchmark of multi-hop ambiguous questions spanning three ambiguity types that are not covered by existing resources. CLARION is also a meaningful contribution: while it is related to standard ReAct-style approaches, our framework differs in that a dedicated **Planning Agent and Acting Agent interact explicitly** to detect, type, and clarify ambiguity before and during reasoning. This ambiguity-aware interaction leads to substantial performance improvements, and our results in Table 4 show that concretizing the Planning modules yields further gains compared to the vanilla ReAct baseline.

---

> > ### Author Response · Authors · 2025-11-27
> >
> > **Q1 – Difficulty of MIRAGE for frontier models.** \
> > As shown in Table 4, models commonly regarded as frontier systems, such as Gemini-2.5 and DeepSeek-v3.1, achieve notably low performance on MIRAGE. This indicates that even **robust large-scale LLMs struggle to resolve the multi-hop ambiguities** present in our benchmark. These results demonstrate that MIRAGE is inherently challenging and effectively exposes limitations in current state-of-the-art reasoning models.
> >
> > **Q4 – Performance by ambiguity type.** \
> > The performance of each method across the three ambiguity types is provided in [TABLE B] below:
> >
> > [TABLE B]
> > |Model|Ambiguity-Type|STR-EM|Disambig-F1|LLM-as-a-Judge|
> > |---|---|:---:|:---:|:---:|
> > |Qwen3-235b|Semantic|0.3390|0.2486|3.303|
> > |Qwen3-235b|Syntactic|0.3500|0.2581|3.370|
> > |Qwen3-235b|General|0.4645|0.3391|3.860|
> > |Gemini-2.5|Semantic|0.2486|0.2361|2.637|
> > |Gemini-2.5|Syntactic|0.2967|0.2653|2.652|
> > |Gemini-2.5|General|0.3279|0.2871|3.059|
> > |Deepseek-v3.1|Semantic|0.3023|0.2511|2.940|
> > |Deepseek-v3.1|Syntactic|0.2933|0.2717|2.840|
> > |Deepseek-v3.1|General|0.3443|0.2878|3.316|
> >
> > Across models and metrics, General ambiguity consistently yields the highest performance, followed by Syntactic and then Semantic ambiguity. This pattern holds across all three evaluation metrics—STR-EM, Disambig-F1, and LLM-as-a-Judge—showing stable and consistent trends.

---

### Official Review · Reviewer_8JCs · 2025-11-03

**Soundness:** 2
**Presentation:** 2
**Contribution:** 2
**Rating:** 4
**Confidence:** 4

**Summary:**

This paper proposes a QA dataset consisting of ambiguous multi-hop queries, combining the two known factors (ambiguity and multi-hop reasoning) that challenge QA systems. The paper furthermore offers a new model that is specifically tailored to this problem, that improves performance over vanilla LLMs and several retrieval-based alternatives. I am generally in favor of small-scale, targeted datasets, and I think the dataset is potentially useful to the community if some issues I raise below turn out to be not serious. The model perhaps less so since it's very specific to the problem at hand, but I think the claimed contribution is for it to be a strong baseline for the proposed dataset, and it serves that purpose well. I have various conceptual issues that holds me back from assigning a higher rating (details below), but maybe this can be resolved through discussions.

**Strengths:**

- The work addresses a dataset gap in the literature (Table 5), and the dataset will be useful for some subset of our community.
- The steps taken to construct the dataset seem generally sensible.

**Weaknesses:**

- The scope of the contribution seems slightly narrow, although this isn't a make-or-break weakness.
- Missing various details about human annotators: e.g., who they are (their qualification and background), recruitment process, how they were compensated, etc. There is some information in the Appendix but seems insufficient.
- Human inter-annotator agreement for human evaluation should also be reported (and maybe also LLM evaluator agreement just out of interest?). I also do feel like 20 questions per type is a bit too few although I appreciate the analysis existing.
- Several conceptual issues:
	- "General ambiguity" seems like a bad term for the type of ambiguity the authors are shooting for. This makes is sound like a catch-all category, although it actually means something very specific about over-constrained queries that lead to a null set answer. I also have doubts about whether this category should really be treated as an ambiguity at all, because this is now based on inferences about the intent of the question above and beyond what is actually stated, and that intent may or may not even be true (I think this might be my confusion with the "city center" question, see below under presentation clarity issues). Maybe the question asker truly intended the narrow scope query and if so, the answer should be "Unanswerable" - I think this is actually how people have treated such queries in prior work.
	- Although not strictly a weakness but more of an untouched issue is the treatment of many-way ambiguities: see Questions..
- Some presentation clarity issues throughout the paper. A few examples:
	- Interpretation of Figure 2: First, I'm not sure if I understand why this is a syntactic ambiguity. The first bifurcation of the interpretation seems to be reference ambiguity of the phrase "a Nobel Chemistry laureate", which is not syntactic ambiguity (this actually makes me question the validity of the ambiguity categories proposed, so maybe this is more than a presentation clarity issue). Second, the sub-question steps don't really make sense to me: the sensical sub-questions route "Which Nobel Chemistry laureate is being referred to?" and then the two questions "Which element discovered by Curie?" and "Which element discovered by Seaborg?", and then finally the actual dates. I also think there's another higher-level ambiguity here where is a version of the question that is not actually ambiguous, which is about the chronologically first instance of an element discovered by a Nobel Chemistry laureate being officially recognized by the international union. Then there is genuinely one answer - the instance that comes first in history. I think this also connects to the issue I will discuss in Questions about multi-way ambiguities.
	- The rationale behind L158-180 was difficult to understand. The query literally says "city center", why should we expect that committing to city center is premature and consider the broader metropolitan area? Is it because it could not be the case that the Olympics being queried about is only held at the city center? (i.e., the assumption here that the absence of an answer to the query requires generalization of the query to find closest valid answer, but I'm not sure if this is justifiable) I'm not sure if this can really be grouped as an ambiguity type, since the ambiguity itself is of the kind that is covered in other categories (the reference ambiguity of "the Olympics"), but what's different is the scope of the query (which is not about ambiguity at all).

**Questions:**

My main questions concern the n of disambiguated questions I have three questions regarding this. (1) is about the statistics/coverage and what the implication of this is for the proposed dataset, (2) is a more general question about what an ideal QA system should be doing, and (3) is a more nitpicky question about the potential of combinatorial explosion.

(1) A question can be n-way (n>=2) ambiguous, as you also note in your definition of clarifying question in L238. What is the average number of n in your dataset? I could see it potentially being quite large especially for questions like your illustrative example in Figure 3. While only two-way ambiguity is shown in the figure, according to Wikipedia page for Paris (https://en.wikipedia.org/wiki/Paris_(disambiguation) it is a LOT more than 2-way ambiguity. Does the dataset cover all of these cases (how can one even check)? If it only offers a partial coverage, isn't this problematic in that it encourages model development that only rewards a specific subset of all possible ambiguity?

(2) In the case of many-way ambiguities like the Paris one, I'm not sure what a QA system even should do. If I, as a user, were asking questions like the one in Figure 3, I clearly don't want to have 40 different answers even if it were truly 40-way ambiguous. I'm curious whether you have any thoughts on this.

(3) A question could technically display more than 1 ambiguity types: e.g., it could be syntactically and semantically ambiguous, or it could exhibit more than one semantic ambiguity. In this case, there could be a combinatorial explosion of ambiguities. Are there cases like this in the dataset and how should we deal with this? I think the general issue I'm getting at is that there needs to be a way to rank "worth considering" ambiguities among many technically possible ambiguities, and there is a clear limitation in only treating ambiguities that multiple LLMs agree on as true ambiguity or ambiguities worth considering.

---

> ### Author Response · Authors · 2025-11-27
>
> **W1-1) Human annotator details.** \
> We were not able to include all details in the main paper due to space limitations, but we now provide a more complete description in Appendix D. In our study, **five graduate-level annotators who are fluent in English** participated in the labeling process. All annotators had prior experience with NLP or IR research and were familiar with the concepts of ambiguity and multi-hop question answering. Each annotator was compensated at a rate of **10 USD per hour**.
>
> **W1-2) Inter-annotator agreement.** \
> As the reviewer correctly noted, reporting agreement among annotators is important for establishing the reliability of our evaluations. Accordingly, we computed **Fleiss’ κ, average pairwise agreement (P̄), strict agreement, and majority agreement** for (1) the long-answer judgment criteria (Relevance, Faithfulness, Informativeness, Correctness) and (2) the dataset-quality assessment criteria (Ambiguity, Clarification, Long Answer). The results are provided below:
>
> | Item           | Fleiss’ κ | P̄    | Strict Agree | Majority Agree |
> |----------------|:-----------:|:-------:|:--------------:|:----------------:|
> | Relevance      | 1.000     | 1.000 | 1.000        | 1.000          |
> | Faithfulness   | 1.000     | 1.000 | 1.000        | 1.000          |
> | Informativeness| 0.907     | 0.978 | 0.967        | 0.989          |
> | Correctness    | 1.000     | 1.000 | 1.000        | 1.000          |
> | Ambiguity      | 0.589     | 0.978 | 0.967        | 0.989          |
> | Clarification  | 0.851     | 0.989 | 0.983        | 0.994          |
> | Long Answer    | -0.006    | 0.989 | 0.983        | 0.994          |
>
> Overall, the results indicate **strong agreement across annotators**. The low κ for Long Answer is a well-known artifact of Fleiss’ κ under extreme class imbalance (prevalence): because nearly all annotators selected the same label (“Y”), the expected chance agreement becomes inflated, which in turn suppresses κ even when actual agreement is very high. This does not reflect genuine annotator disagreement. The high strict and majority agreement values further verify that annotators were **highly consistent** for this and all other criteria.
>
>
> **W1-3) Number of samples for human annotation.** \
> This task is inherently demanding. Our pilot study showed that annotating **60 samples required approximately five hours per annotator** due to the need to inspect the original ambiguous question, verify the type of ambiguity, check whether the clarification steps genuinely disambiguate the question, and ensure that the long answer faithfully integrates all short answers. Given the time and cost required per sample, we determined that the current number of annotated instances represents a realistic and appropriate scale within the scope of this work.
>
>
> **W2-1) Interpretation of Figure 2.** \
> We have revised Figure 2 to more clearly highlight that the core ambiguity in the question “How did the restrictions on foreign banks affect the expansion that followed?” arises from differences in the semantic attachment scope of the term “expansion”. Specifically, the ambiguity stems from whether “expansion” is interpreted as referring to **GDP-level economic expansion or to banking-sector expansion**. When “expansion” is attached at the GDP level, the question is naturally understood as asking about a two-hop, economy-centered reasoning chain: how the restrictions on foreign banks influenced the overall economy and, in turn, the subsequent trajectory of GDP growth. In contrast, when “expansion” is attached at the banking-sector level, the question instead triggers a finance-centered reasoning chain: how the restrictions affected banks’ ability to expand and how this shaped the later growth of the financial sector. Thus, even though the surface form of the question is identical, a **difference in attachment scope alone leads to entirely different multi-hop reasoning chains**. The revised figure does not explicitly depict a full parse tree; rather, it focuses on (i) the two clearly distinct sub-question chains induced by the alternative attachments, (ii) the blended reasoning that arises when an LLM fails to keep these branches separate and merges elements from both, and (iii) the resulting failure mode in which one branch is effectively treated as a Missing Answer. In the main text, we clarified that this example is a case of semantic attachment ambiguity grounded in structural differences in attachment.

---

> ### Author Response · Authors · 2025-11-27
>
> **W2-2) Sub-question steps.** \
> The purpose of Figure 2, however, is not to propose the single “best” decomposition, but rather to **visualize a real failure pattern** we observed in LLMs: the model interprets the sentence in one structural way, commits early to a single reasoning branch, and consequently misses the alternative branch entirely. In the revised text, we explicitly state that the sub-question chain shown in Figure 2 is intended to reflect an **empirically observed failure mode**—where the model prematurely collapses an inherently ambiguous structure—rather than to prescribe the ideal decomposition strategy.
>
> **W2-3) Rationale behind L158–180.** \
> The example in this section was intended to illustrate not a simple entity ambiguity, but a case of an over-constrained query where the surface formulation of the question is excessively narrow and does not adequately reflect the underlying information need. In MIRAGE, general ambiguity refers to situations in which a question becomes **overly fine-grained or unnecessarily restrictive**, causing retrieval and reasoning to diverge from the user’s actual intent. The requirement to provide the year the Olympics were held, specifically in the city center, rather than in the host city more broadly, is a representative case that produces a scope mismatch between the user’s intent and the literal constraints stated in the question.
>
> **W3) - Concerns about the scope of the term general ambiguity.** \
> In our work, we use this term to refer to cases where the question is over-specified relative to the user’s underlying intent, such that the stated constraints are excessively narrow and do not fully capture the actual information need. In these situations, it becomes unclear what the questioner genuinely aims to ask, and multiple reasonable interpretations can coexist. We therefore treat this phenomenon as a distinct form of ambiguity.
>
> **Q1) – Question about multiple interpretations.** \
> MIRAGE includes questions with two or more plausible interpretations, and in practice, each item contains an average of 2–3 distinct readings. Some questions indeed allow for more than three interpretations, as in the provided example, but we focus only on the **major interpretations** that are verifiable based on Wikipedia evidence.
>
> **Q2) – What should a QA system do under many-way ambiguity?** \
> In many-way ambiguity scenarios—for example, as the reviewer mentions, when a single question can in principle admit up to 40 different interpretations—a QA system should, at least normatively, be aware of the **full space of possible readings**. However, in real-world settings, such extreme many-way ambiguity is not an artificial corner case, but rather a natural manifestation of the long-tail distribution inherent in real data. MIRAGE is **built directly from the empirical distributions** of Wikipedia and MuSiQue, so it includes not only frequently occurring ambiguity patterns, but also rare edge cases where the combinatorial complexity of interpretations grows rapidly. In this sense, many-way ambiguity is not an anomaly to be filtered out, but an intrinsic property of the information-seeking process that models will inevitably encounter in practice.  Ideally, the system should enumerate all possible interpretations, but in reality, many questions have only a few interpretations, while a very small number of long-tail questions branch into dozens, and MIRAGE is designed to **follow this distribution itself**.
>
> **Q3) – Prioritization when multiple ambiguity types coexist in a single question.** \
> During dataset construction, we identified some cases where a question exhibited more than one ambiguity type simultaneously. However, such instances make it **difficult to clearly describe the underlying source of ambiguity** and can introduce ambiguity into the evaluation criteria themselves. For this reason, we **excluded all multi-type cases** and included only questions with a single, clearly identifiable ambiguity type in the final dataset.

---

### Author Response · Authors · 2025-11-27

We would like to kindly inform you that all revisions addressing the reviewers’ comments have been **highlighted in blue** throughout the revised manuscript. We appreciate your time and consideration in reviewing these changes.

---

### Note · Authors · 2025-12-03

I have read and agree with the venue's withdrawal policy on behalf of myself and my co-authors.